# An analysis modality for vascular structures combining tissue-clearing technology and topological data analysis

Kei Takahashi [1], Ko Abe[2,10], Shimpei I. Kubota [1,10], Noriaki Fukatsu [3,10], Yasuyuki Morishita[1], Yasuhiro Yoshimatsu[4], Satoshi Hirakawa[5], Yoshiaki Kubota [6], Tetsuro Watabe[7], Shogo Ehata [1], Hiroki R. Ueda[8,9], Teppei Shimamura[3] ✉ & Kohei Miyazono [1] ✉

The blood and lymphatic vasculature networks are not yet fully understood even in mouse because of the inherent limitations of imaging systems and quantification methods. This study aims to evaluate the usefulness of the tissue-clearing technology for visualizing blood and lymphatic vessels in adult mouse. Clear, unobstructed brain/body imaging cocktails and computational analysis (CUBIC) enables us to capture the high-resolution 3D images of organ- or area-specific vascular structures. To evaluate these 3D structural images, signals are first classified from the original captured images by machine learning at pixel base. Then, these classified target signals are subjected to topological data analysis and non-homogeneous Poisson process model to extract geometric features. Consequently, the structural difference of vasculatures is successfully evaluated in mouse disease models. In conclusion, this study demonstrates the utility of CUBIC for analysis of vascular structures and presents its feasibility as an analysis modality in combination with 3D images and mathematical frameworks.

The blood and lymphatic vasculatures maintain fluid homeostasis in the animal body, and their functions are closely related to various types of diseases including cancer. The blood vasculature is a closed circulatory system that transports oxygen and nutrients and consists of arteries, veins, and capillaries[1]. On the other hand, the lymphatic vasculature is a one-way network with thin-walled capillaries that transports lymph, cholesterol, and immune cells. It is comprised by lymphatic vessels, lymph nodes, and associated lymphoid organs such as the spleen and thymus[1,2]. These anatomical structures have thus far been studied mainly by two-dimensional (2D) immunohistochemistry (IHC).

Recent imaging modalities, such as computed tomography (CT), magnetic resonance imaging (MRI), optical imaging with near-infrared fluorescence dyes, and ultrasound array-based real-time photoacoustic microscopy, have enabled the visualization of the blood/lymphatic vasculature with three-dimentional (3D) images[3,4]. Two-photon microscopy and confocal microscopy are also utilized for monitoring the structures of capillaries in mouse[5,6]. Thus, imaging technologies have been improved for visualization of the vasculature; however, the structures of the vasculatures, especially of lymphatic vessels, are still not fully understood to date due to the inherent limitations of imaging systems. In this regard, a better and more appropriate imaging approach with high resolution is needed to fully understand the vasculature network at the entire organ level.

The development of tissue-clearing technology in the previous decades has led to the introduction of numerous tissue-clearing methods that are currently mainly used in neuroscience research[7]. These methods enable the capture of high-resolution 3D images without sectioning or pulverization of mouse organs[8]. Recently, tissue-clearing approaches started to be used not only for neuroscience research, but also for analysis of diseases including cancer[9]. We previously utilized the hydrophilic chemical-based method called clear, unobstructed brain/body imaging cocktails and computational analysis

(CUBIC) for cancer research[10–12]. CUBIC reagents contain cocktails selected by chemical screening[13]. In those previous studies, the CUBIC system was useful for the detection of cancer cells with single-cell resolution at the whole mouse body/organ level[10,13]. We thus far focused on visualization of cancer cells to detect metastatic tumors even in the deeper tissues. However, not only cancer cells, but also stromal cells in surrounding microenvironment should be investigated to understand the mechanism of cancer metastasis or drug resistance[14,15]. Since vascular dysfunction is also associated with various diseases, visualization of the vasculatures with the CUBIC system may be the next challenge to overcome. Moreover, 3D imaging with the CUBIC system is suitable for analysis of complex vascular structures in vivo. Indeed, some studies recently reported the use of tissue-clearing methods for visualizing blood and lymphatic capillaries in mouse[16–23].

In addition to the limitations of imaging systems for analysis of vasculatures, no standard method for evaluating the 3D images of the whole mouse organ has been established. For the evaluation of vascular structures, quantifying their volumes or counting branching points based on 2D images is still a standard way. Therefore, we decided to establish an evaluation approach in combination with mathematical frameworks, called topological data analysis (TDA) and non-homogeneous Poisson process (NHPP). TDA broadly means the statistical methods that find structures in data[24] and includes various methods, e.g., clustering, manifold estimation (learning), ridge estimation, and persistent homology (PH)[24]. It has been applied to biological studies such as viral reassortment or cancer genomics[25,26]. Moreover, PH is used in the analysis of brain artery with magnetic resonance angiography images[27]. NHPP is a Poisson point process that has variable intensity in the domain in which it is defined and commonly used in a wide range of applications, for example, when modeling the failures of repairable systems[28], the occurrence of earthquakes[29], or the evolution of customer purchase behavior[30]. However, there were no studies on the application of TDA and NHPP in combination with tissue-clearing technology.

This study aims to evaluate the usefulness of the tissue-clearing technology CUBIC for the visualization of blood and lymphatic vessels in adult mouse. Towards this goal, geometric features were extracted using PH[31]. The main advantage of TDA is that geometric features can be extracted without any prior definition of geometric information, such as container length, volume, shape, network or tree, spatial distribution, or density of vessels. In addition to TDA, NHPP is also used to model blood vessel intensity occurrence, which allows for inter-individual and inter-site comparison of vessel intensities.

## Results

### Visualization of blood vessels in various mouse organs

We used the CUBIC protocol to visualize the blood and lymphatic vessels in adult mouse in the present study. The basic protocol with CUBIC reagents (CUBIC-L and CUBIC-R (N)) includes three steps, i.e., fixation with paraformaldehyde (PFA) (1 day), delipidation with CUBIC-L (2-7 days), and refractive index (RI) adjustment with CUBIC-R (N) (1–2 days) (Fig. 1a). In case of 3D immunostaining, samples were incubated with antibodies for 3-4 days. Some thin organs, whose images were hardly captured in observation oil (e.g., the skin), were embedded in 2% gelatin before RI adjustment as previously described (Fig. 1a)[32]. This CUBIC protocol makes us possible to visualize the whole mouse body or organ with single-cell resolution[10]. Blood vessels in mice were visualized by injecting Cdh5-BAC-Cre^ERT2; ROSA-lox-stop-lox-tdTomato (VE-cad-tdTomato) mice intraperitoneally with tamoxifen (Tx) to induce the expression of Cre recombinase in vascular endothelial (VE)-cadherin-positive (VE-cad+) endothelial cells, leading to the expression of tdTomato fluorescent protein. These mice were utilized to visualize blood capillaries with not only 2D but also 3D images in mouse embryos or newborn mouse babies without tissue-clearing methods[33]. However,

in adult mice, there are few reports showing the 3D visualization of blood capillaries, because of the lack of monitoring system. After Tx injection, adult mice were sacrificed and subjected to CUBIC procedures as shown in Fig. 1a. We successfully visualized blood capillaries at the whole mouse level (Fig. 1b). Signals from a VE-cad-tdTomato mouse were stronger and clearer than those from samples from a normal mouse (Supplementary Fig. 1a). The enlarged images clearly showed condensed blood capillaries in the liver, stomach, and intestine. The blood capillaries of each excised organ (brain, lung, stomach, spleen, heart, intestine, kidney, and liver) were also visualized using this transgenic mouse (Fig. 1c and Supplementary Fig. 1b). Enlarged images of the stomach and kidney and the 2D images of each organ (spleen, kidney, and intestine) are shown in Fig. 1d, e, respectively. The 2D images of the intestine showed that each villus has VE-cad+ blood capillaries (Fig. 1e), while the 2D image of the spleen showed the existence of high-density blood capillaries except in white pulp areas (Fig. 1e). The 3D and 2D images of the kidney clearly showed each glomerulus and extended smaller arteries (Fig. 1c–e and Supplementary Movie 1). These images were so clear that we can count the number of glomeruli per kidney.

Lectin inoculation enables the visualization of blood vessels in the CUBIC system[34]. Thus, the mice were injected with tomato-lectin conjugated with Texas-Red before they were sacrificed. As shown in Fig. 1f, the images of the kidney in lectin-injected mice were similar to those in VE-cad-tdTomato mice. Both images in VE-cad-tdTomato mice and lectin-injected mice showed blood capillaries in the glomeruli. The image of the intestine in lectin-injected mice also showed that blood capillaries exist in each small villus (Fig. 1f). To monitor the structure of blood capillaries in the skin and peritoneum, the samples were embedded in 2% gel. The blood vessels in the skin had a more condensed density than those in the peritoneum (Fig. 1g). To simultaneously visualize the mature blood vessels and blood capillaries, α-smooth muscle action (α-SMA) staining was performed as previously described[35,36] and all blood vessels including the capillaries were monitored in VE-cad-tdTomato mice. The result showed that VE-cad+ blood capillaries were running through the entire heart, while α-SMA enabled visualization of specific large arteries including the coronary arteries (Supplementary Fig. 1c). Two-dimensional IHC for signal confirmation showed that the signals from lectin (conjugated with fluorescein isothiocyanate [FITC])-injected mice overlapped the signals from VE-cad-tdTomato mice (Supplementary Fig. 1d). Moreover, the CD31+ area in the brain overlapped the signals from VE-cad-tdTomato and the signals from lectin (conjugated with Texas-Red)-injected mice (Supplementary Fig. 1d). From these data, blood capillaries were successfully visualized at whole organ level with high resolution in VE-cad-tdTomato adult mice using CUBIC imaging.

### Visualization of lymphatic vessels in various mouse organs

Next, we tried to visualize lymphatic vessels in various organs using a Prox1-green fluorescent protein (GFP) mouse, which is a transgenic mouse expressing GFP under the Prospero homeobox transcription factor 1 (Prox1) promoter[37]. Prox1 is a transcription factor that plays key roles in lymphatic development that is expressed in lymphatic endothelial cells[38]. Previous reports have already shown the running of lymphatic vessels using these Prox1-GFP mice, especially in the brain[39]. However, data are still not enough, especially for 3D structure analysis. To enhance the signals from GFP, samples were stained with anti-GFP antibody. Our results revealed varying structural patterns of lymphatic vessels at the whole organ level, including in the lung, brain, intestine, heart, kidney, stomach, pancreas, and eye (Fig. 2a, b). In the brain, it has been already reported that lymphatic vessels run parallel to the dural sinus[39,40]. Our result supports this work and showed that Prox1+ lymphatic vessels run along with not only the superior sagittal sinus, but also the transverse sinus and ventricle

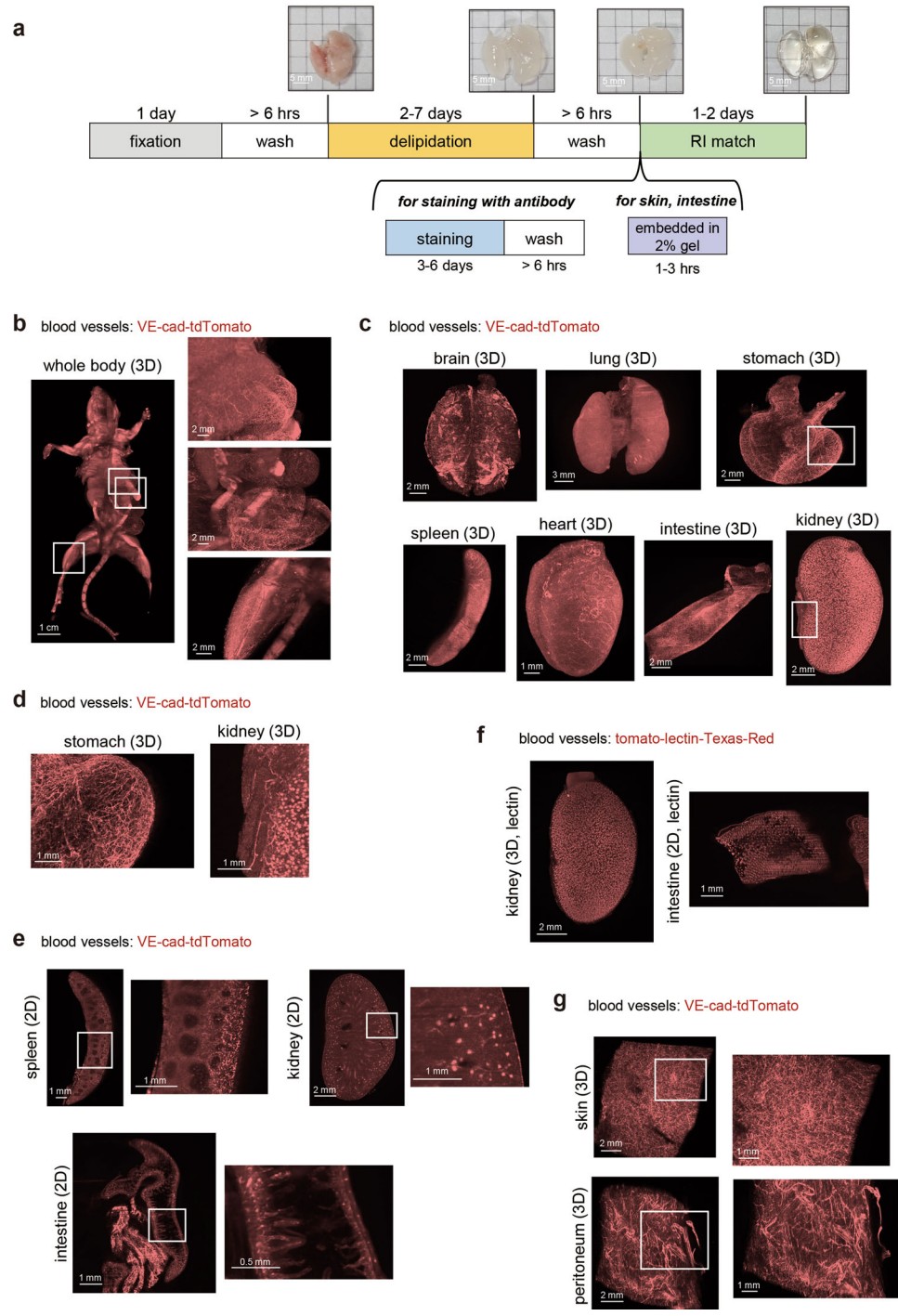

**Fig. 1 | Whole-body and whole-organ 3D imaging of mouse blood vessels.**
**a** Whole-body/organ tissue-clearing protocol with CUBIC reagents. Mice were sacrificed and perfused with 4% PFA. The fixed samples were subjected to CUBIC procedures. The basic protocol consists of fixation, delipidation with CUBIC-L, and RI adjustment with CUBIC-R (N). For 3D immunostaining, staining with washing is inserted before RI adjustment. Samples from the skin and peritoneum were embedded into 2% gelatin before RI adjustment. The images of the transparent samples are captured using light sheet fluorescent microscopy (LSFM). The images of the lung in each step are shown. **b** The 3D-reconstituted (3D) body image of a VE-cad-tdTomato mouse (female, 4 months). Before CUBIC procedures, tamoxifen (Tx) was injected (i.p.) for the induction of reporter. The enlarged images of the abdominal organs (liver, stomach, intestine and leg) are shown in the right panels.

**c** The 3D images of the whole organs, including brain, lung, stomach, spleen, heart, intestine, and kidney in VE-cad-tdTomato mice (2–4 months) (Z; 10 μm step, digital zoom; brain, lung: 1.25, stomach, spleen, and kidney: 1.6, heart: 2.5, intestine: 2.0). **d** The enlarged 3D images (white insets) (kidney and stomach) of (**c**). **e** The 2D (XY) images of the spleen, kidney, and intestine. White insets are magnified next to each image. **f** The 3D image of the kidney and 2D image of the intestine in tomato-lectin-injected mice (C57BL/6J, 5w). The mouse was sacrificed 5 min after inoculation of Texas-Red conjugated with tomato-lectin (Z; 10 μm step, digital zoom; kidney: 2.0, intestine: 1.6). **g** The 3D images of the blood vessels in the skin and peritoneum embedded in gelatin. The skin and peritoneum samples from a VE-cad-tdTomato mouse (3 months) were embedded in 2% gel before RI adjustment. White insets are magnified next to each image (Z; 10 μm step, digital zoom; 2.0).

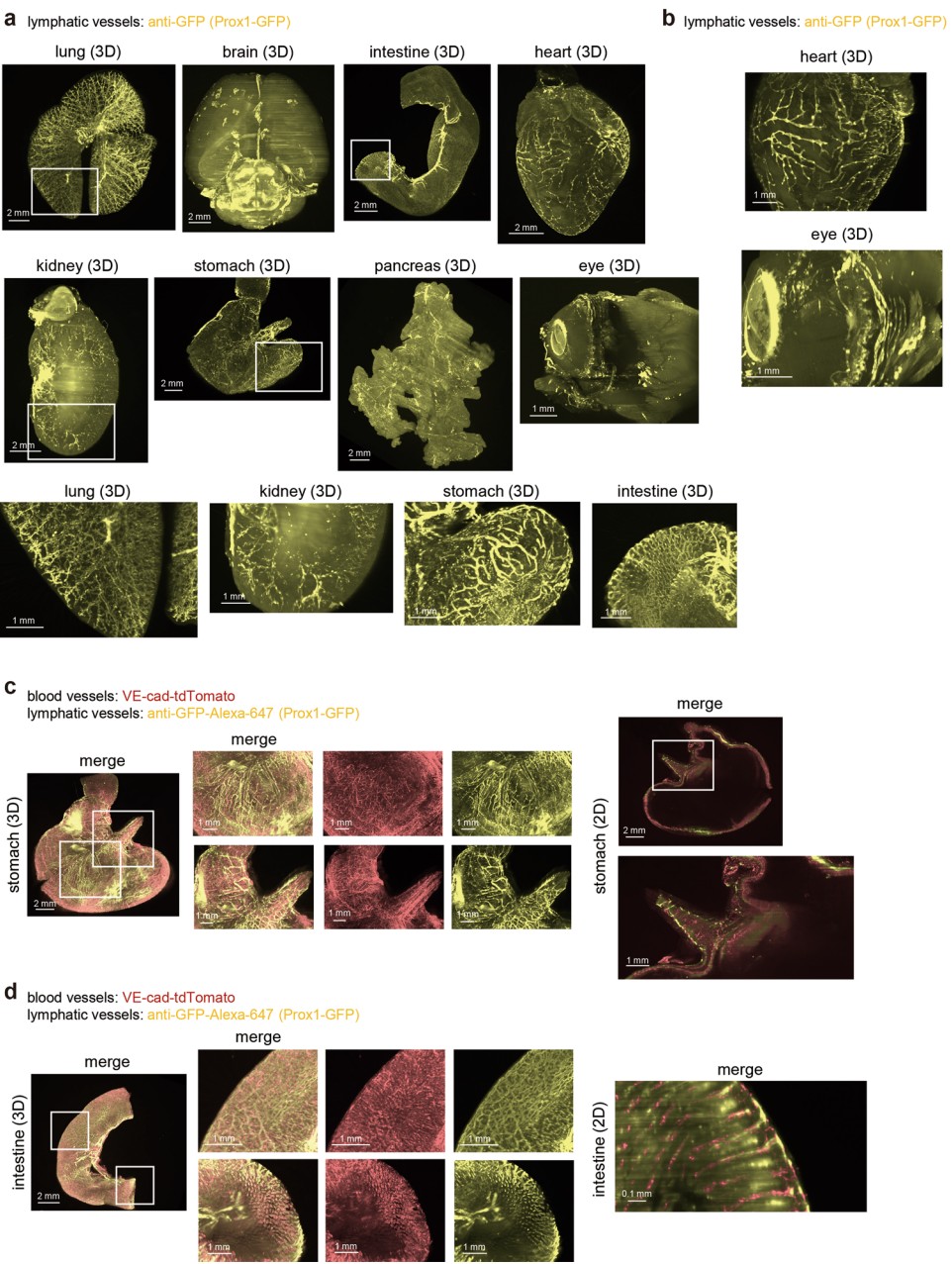

**Fig. 2 | Whole-organ 3D imaging of mouse lymphatic vessels. a** The 3D whole-organ images of mouse lymphatic vessels in the lung, brain, intestine, heart, kidney, stomach, pancreas, and eye. Prox1-GFP mice (2-4 months) were sacrificed and subjected to CUBIC procedures as shown in Fig. 1a. To enhance the signals from Prox1-GFP, samples were stained with anti-GFP antibody ($Z = 10$ μm step, digital zoom; lung, brain, stomach, and pancreas 1.25, intestine and kidney: 1.6, heart: 2.0, eye: 4.0). The enlarged 3D images (white insets) are shown in the bottom. **b** The

enlarged 3D images of the heart and eye shown in (**a**). **c, d** Simultaneous visualization of blood and lymphatic vessels in the stomach and intestine. VE-cad-tdTomato-Prox1-GFP mice (2-4 months) were sacrificed, and samples were subjected to CUBIC procedures. The 3D whole-stomach (**c**) and intestine (**d**) images are shown. The enlarged images of white insets are shown next to whole-organ images. The 2D images (XY) are also shown in the right panel ($Z = 10$ μm step, digital zoom; stomach: 1.25, intestine: 1.6 and 6.3).

(Fig. 2a). In the eye, Prox1 was expressed not only in lymphatic vessels in corneal limbus, but also in the lens and Schlemm's canal, as previously reported based on IHC section data (Fig. 2b)[41]. In the heart, lymphatic vessels covered both right and left sides of the heart, and the epicardial lymphatics were well visualized (Fig. 2b). In the stomach, the lymphatic vessels in the forestomach showed branched-like structures, whereas lymphatic vessels running in the glandular stomach (in human: corpus) were directional and straight. Meanwhile, lymphatic vessels in the lung went along with the bronchi (Supplementary Movie 2), and those in the kidney went along with arteries and the nerves. Small lymphatic capillaries existed in the center of each villus in the intestine. We also embedded samples

from the intestine and mesentery in gelatin and visualized lymphatic vessels in these tissues (Supplementary Fig. 2a). These data suggested organ- or area-specific structures of lymphatic vessels.

Lymphatic vessel endothelial hyaluronan receptor 1 (LYVE1) and vascular endothelial growth factor receptor 3 (VEGFR3/FLT4) are representative markers of lymphatic vessels. Lymphatic vessels in the stomach and intestine were also visualized using 3D immunostaining with anti-VEGFR3 antibody (Supplementary Fig. 2b)[10]. The result of 3D immunostaining with anti-VEGFR3 antibody in Prox1-GFP mice showed that the signals from Prox1-GFP overlapped with, but gave clearer images than, those with VEGFR3 staining (Supplementary Fig. 2c).

For signal confirmation, IHC was performed using anti-LYVE1 antibody in Prox1-GFP mice. The result showed that the LYVE1 staining overlapped the signals from Prox1-GFP in the lung (Supplementary Fig. 2d), and the lymphatic vessels in the intestine were also visualized using 2D IHC with anti-LYVE1 antibody (Supplementary Fig. 2e). We also generated VE-cad-tdTomato; Prox1-GFP mice by crossing VE-cad-Cre-tdTomato mice with Prox1-GFP mice to simultaneously visualize both blood vessels and lymphatic vessels. Interestingly, we found that lymphatic and blood capillaries did not go along with each other in the stomach, and the structures of these capillaries differed between the forestomach and glandular stomach (Fig. 2c). Further, lymphatic and blood capillaries were observed in each intestinal villus (Fig. 2d). These findings clearly showed that lymphatic vessels existed in the center of villus and were surrounded by blood capillaries (Fig. 2d), supporting previous observations about the anatomical structures of lymphatic and blood vessels.

## Quantification of the visualized 3D images of capillaries

Present findings showing the 3D images by CUBIC imaging were so clear that we could capture the precise structure of the blood and lymphatic vasculatures at the whole organ level. However, auto-fluorescence signals were often a problem, and signals were sometimes too low to observe. In addition, there is no appropriate quantification method to show the structural difference observed in various 3D images. Therefore, we decided to establish an analysis method using mathematical frameworks.

Tiff images with original signal intensities were captured by light sheet fluorescent microscopy (LSFM), and these tiff images were converted into hdf5 files with python. Signals were classified using ilastik software as previously described[42–44] (Fig. 3a). The probabilities of signal intensities were saved as an hdf5 file. These hdf5 files were analyzed by ilastik software and probabilities were exported as an h5 file. To confirm whether the pixel classification works or not, the probabilities were analyzed using Imaris software (Bitplane AG, Zurich, Switzerland) (Supplementary Fig. 3) as previously described[44]. The thresholds for the probability of classified signals were decided manually by comparing the original and classified signals side by side. The structures of the lymphatic capillaries were extracted, as shown in Fig. 3b. This procedure enabled us to cut autofluorescence signals and investigate the positive signals more clearly. Regarding renal lymphatics, extracted signals seem to go along with not only the nerve, but also the artery and vein. In the visualization of both α-SMA⁺ mature blood vessels and VE-cad⁺ blood capillaries in the mouse brain, target signals were also successfully classified with ilastik, as shown in Fig. 3c, d.

## Analysis of α-SMA⁺ mature blood vessels and VE-cad⁺ blood capillaries in the brain

To obtain more information about observed signals of blood vessels in the brain, anatomical annotations in the CUBIC-Atlas (http://cubic-atlas.riken.jp/) were added to them as previously described[32,45,46]

(Fig. 4a). The classified pixel-based signals were analyzed as points using NHPP, which can evaluate both strength ($a$) and directionalities ($b_x$, $b_y$ and $b_z$) of vascular structures based on the point density (Fig. 4a). To compare the capillary expansion between α-SMA⁺ mature blood vessels and VE-cad⁺ blood capillaries in the brain, the Fisher's exact test was used. The result showed that the blood vessel strength (a) of α-SMA⁺ signals in the thalamus (TH) were higher compared to those of VE-cad⁺ signals (Fig. 4b, Supplementary Fig. 4). α-SMA⁺ blood vessels showed the higher directionality in the isocortex (ISO), especially in the X- and Y- (X-/Y-) directions (Fig. 4b), which might include superior sagittal sinus. In addition, α-SMA⁺ blood vessels tended to expand from the midbrain (MB, Y-) and pons (P, X-/Z-) (Fig. 4b), surrounded by arteries including the posterior cerebral artery[47,48]. From

these results, we can speculate that thick α-SMA⁺ blood vessels existed on the surface of the isocortex and boundary of the midbrain to the pons and medulla.

This analysis was repeated with four independent samples (Supplementary Fig. 4) and confirmed by the original 3D images shown in Fig. 4c. The α-SMA⁺ mature blood vessels existed along the midbrain to the brainstem, whereas the VE-cad⁺ blood capillaries did not (Fig. 4c, Supplementary Fig. 5, Supplementary Movie 3). In addition, α-SMA⁺ mature blood vessels covered the isocortex in X- or Y-directions, while VE-cad⁺ blood capillaries uniformly covered the isocortex (Fig. 4c, Supplementary Movie 3). The images of blood vessels were slightly faint, but analysis using NHPP clearly demonstrated the regional difference in cerebral blood vessels. Notably, analysis of blood vessels at the subarachnoid space may be clinically important because it is where bleeding is often observed.

To investigate the diversity of vascular structures in the α-SMA⁺ mature blood vessels or VE-cad⁺ blood capillaries, the geometric features extracted with PH (a main method of TDA) in each brain area were evaluated by the Sliced Wasserstein kernel. Using this method, even if the volume of vasculatures, the number of branch points, and the length among branch points are the same but the structure is different, PH can detect its difference, as shown in Fig. 5a. Therefore, PH is an ideal methodology, which can comprehensively evaluate the vascular structures by using new parameters ($r$) from each target point. Their brain area objects were plotted by a two-dimensional plot that displays the relative positions of the brain area objects from the distances between them using multidimensional scaling (MDS) (Fig. 5a). Using PH in Fig. 5, vascular structures in different areas were compared using each marker, instead of comparing α-SMA⁺ and VE-cad⁺ blood vessels. The results indicated that both α-SMA⁺ mature blood vessels and VE-cad⁺ blood capillaries showed unique expansion patterns in the isocortex and the fiber tracts (fiber) (Fig. 5b, c). In addition, α-SMA⁺ mature blood vessels showed slightly different patterns in the cerebellum (CB) where arteries are running, such as in the superior cerebellar artery and posterior inferior cerebellar artery (Fig. 5b). Most of the α-SMA⁺ mature blood vessels showed the wide and directional vessels where the arteries are running, and VE-cad⁺ blood vessels covered the entire areas with condensed capillaries (Figs. 4c, 5d, e and Supplementary Movie. 3). On the other hand, both the α-SMA⁺ blood vessels and the VE-cad⁺ blood capillaries were running in a certain direction (between surface of the isocortex and center of the brain) in the isocortex, showing the featured structures compared to other brain areas (Figs. 4c, 5d, e and Supplementary Movie. 3). Thus, PH findings highlighted that the difference in each vascular expansion depends on the brain area. We also examined the existing parameters including branching points using TubeMap (Supplementary Fig. 6a)[20]. The result of brain vessels showed that the vertex and edge numbers of VE-cad⁺ blood vessels were higher than those of α-SMA⁺ blood vessels (Supplementary Fig. 6b). The vertex and edge numbers in the isocortex were prominently higher than those in the other brain areas (Supplementary Fig. 6b). Moreover, the edge radii of α-SMA⁺ blood vessels in the isocortex area were larger than those of VE-cad⁺ blood vessels (Supplementary Fig. 6b). These results supported our TDA and NHPP data and showed that the structure of blood vessels in the isocortex area is unique compared to the other brain regions.

To evaluate the effect of thresholds of classified signals on the result of PH, the lung data in Supplementary Fig. 3 with different thresholds were analyzed by PH (Supplementary Fig. 7). The result suggested that the threshold for the probability of classified signals has a certain amount of tolerance, although this manual process should be automated to reduce the bias in the future.

## Structure of lymphatic vessels in pulmonary fibrosis model

These analyses were applied to a mouse lung fibrosis model induced by bleomycin instillation. We chose this model because it is suitable

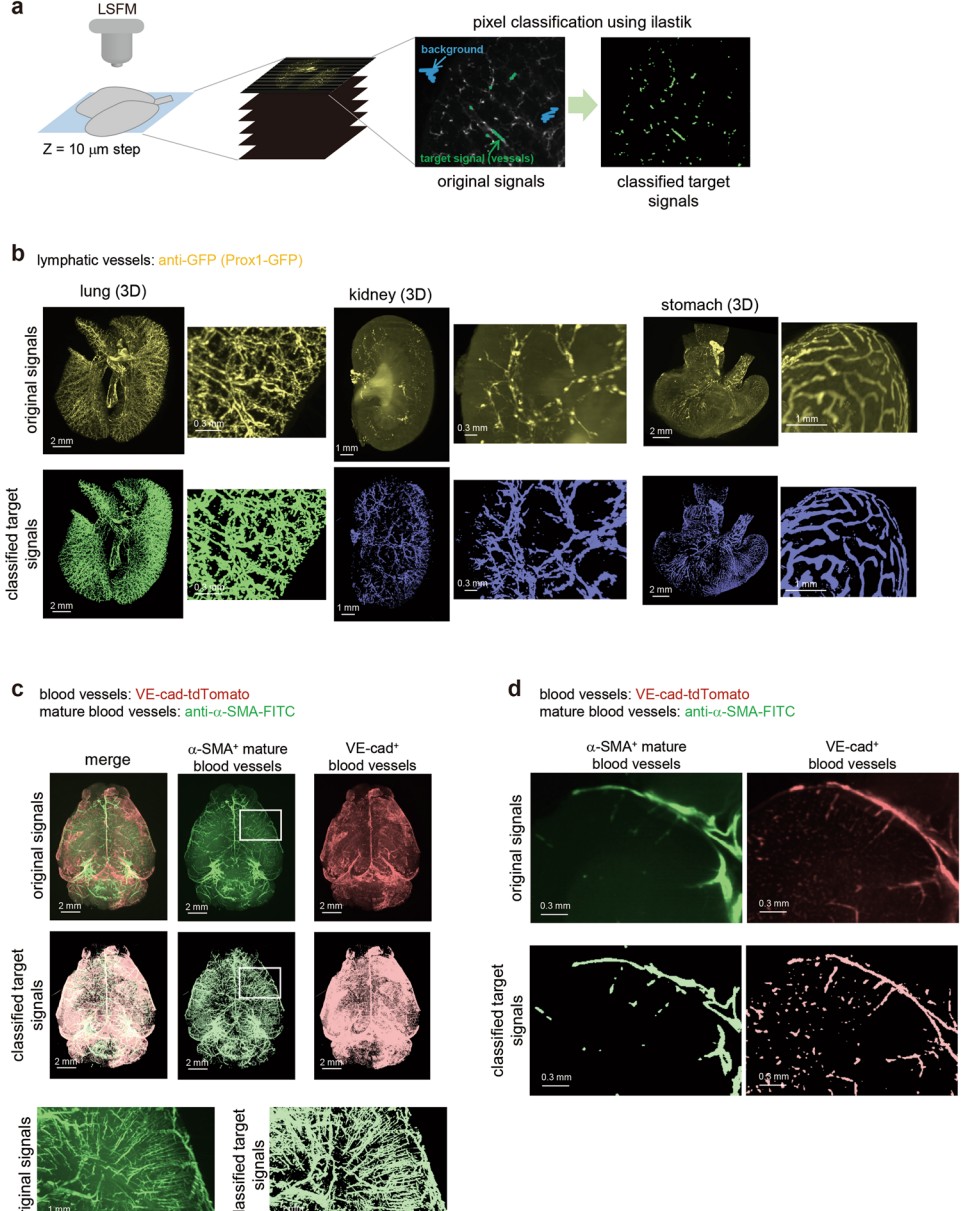

**Fig. 3 | Evaluation of 3D images with computational training. a** Overview of analysis procedure of pixel classification. The 3D images of transparent samples were captured by LSFM. The captured tiff images were converted to hdf5 format using python. Target signals such as vasculature or cancer cells were classified by machine training with ilastik software. In this study, two classes (target signal and background) were defined for the classification. **b** The images of the original signals and classified target signals after training by ilastik. The 3D whole-organ images of the original fluorescent signals (upper) and classified signals after machine learning (lower) are shown (lung, kidney, and stomach). The sample of

the lung is originated from control#1 in Fig. 7. The enlarged 3D images are shown next to whole-organ images. **c** and **d** The 3D and 2D (XY) images of original and classified signals of the blood vessels in the brain. The 3D whole-brain images of α-SMA⁺ mature blood vessels (middle), VE-cad⁺ blood capillaries (right) and their merge images (left) are shown (**c**). Both images with original signals (upper) and classified signals (lower) are shown. The enlarged 3D images of α-SMA⁺ mature blood vessels in the white insets are shown (bottom). The 2D (XY) images are also shown (**d**). Both images with original signals (upper) and classified signals (lower) are shown.

for investigating the structure of lymphatic vessels in lungs damaged by inflammation. Intratracheal bleomycin administration induced severe inflammation (Fig. 6a). To investigate the structure of lymphatic vessels in pulmonary fibrosis, Prox1-GFP mice were intratracheally administered with bleomycin. The images were captured by LSFM and analyzed using Imaris software. Compared to lymphatic vessels in untreated mouse or saline-treated mouse, the structures of the lymphatic vessels, particularly GFP⁺ lymphatic endothelial cells, in bleomycin-treated mice were severely damaged (Fig. 6b). The results from 2D IHC also showed that the structure of lymphatic vessels, especially small ones, was distorted in some bleomycin-

treated mice compared to that in saline-treated mice (Supplementary Fig. 8). Lymphatic vessel signals were classified using machine learning with ilastik. Total classified signals of lymphatic vessels, which can be evaluated as their volume, indicated the reduced signals in some samples in the saline treatment group, and more marked reduction in those of the bleomycin treatment group (Fig. 6c). In addition, analysis of the number of branching/end points, and lengths and radii of branches indicated that the lymphatic vessel structures in the bleomycin-treated lungs were significantly different from those in the control lung (Fig. 6c). The classified signals as lymphatic vessels were also applied to PH to examine the geometric

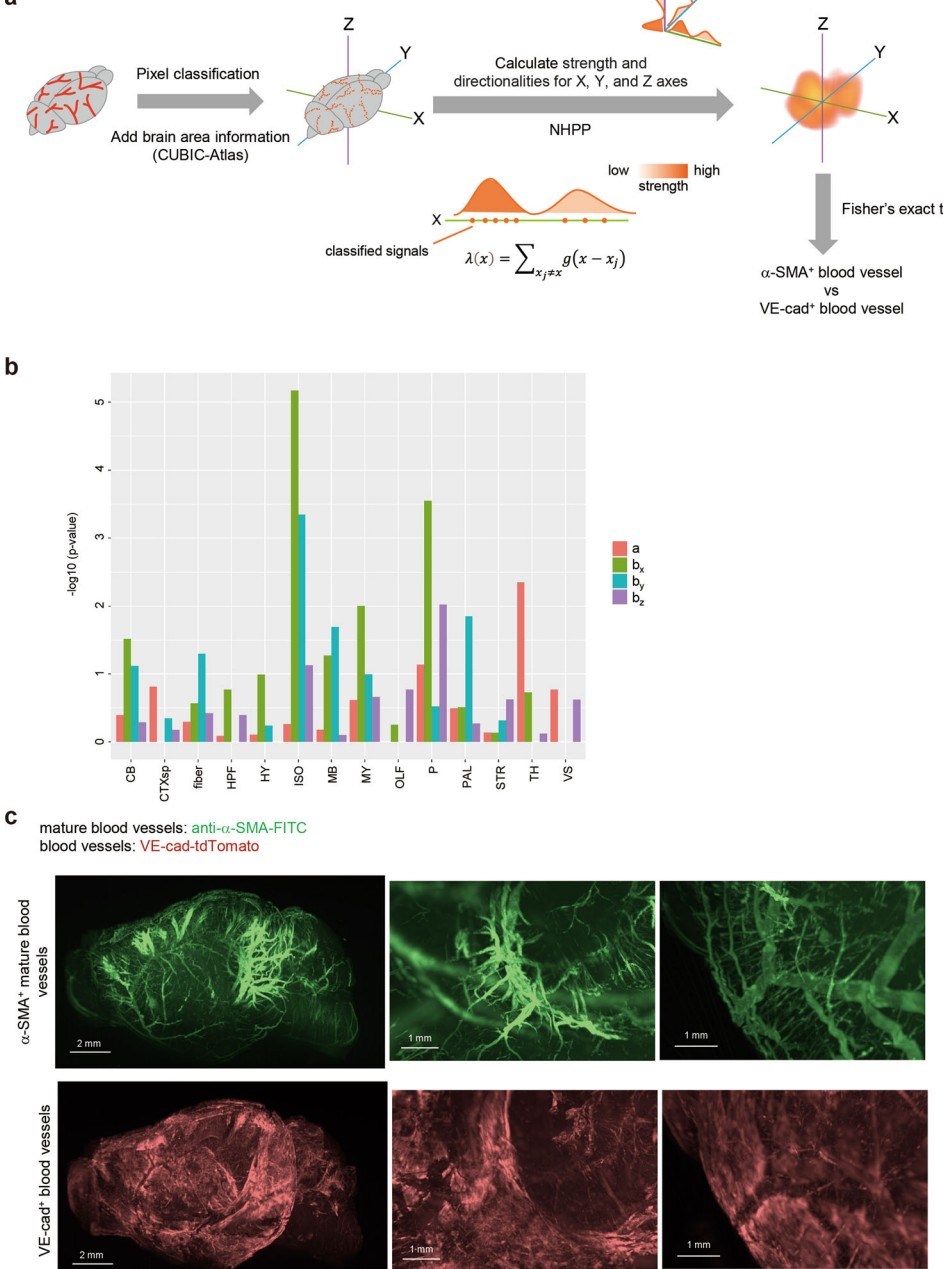

**Fig. 4 | Analysis of brain blood vessels using NHPP. a** Overview of analysis procedure using CUBIC-Atlas and NHPP. First, signals of blood vessels were classified using ilastik software, then the brain anatomical information is added to them using ANTs and CUBIC-Atlas. To evaluate the intensities of blood vessels, non-homogeneous Poisson process (NHPP) was applied. In this method, strength (a) and directionalities ($b_x$, $b_y$, $b_z$) can be calculated using the pixel-based classified signals as the points. To compare the difference between α-SMA$^+$ mature blood vessels and VE-cad$^+$ blood capillaries, Fisher's exact test (two-sided) was used. **b** The result of Fisher's exact test between α-SMA$^+$ mature blood vessels and VE-cad$^+$ blood capillaries in mouse brain. Strength (a) and directionalities ($b_x$, $b_y$, $b_z$) were calculated using NHPP and the results of NHPP in each brain area were used in the Fisher's exact test (14 brain areas: cerebellum (CB), cortical subplate (CTXsp), fiber tracts (fiber), hippocampal formation (HPF), hypothalamus (HY), isocortex (ISO), midbrain (MB), medulla (MY), olfactory areas (OLF), pons (P), pallidum (PAL), striatum (STR), thalamus (TH), and ventricular systems (VS)). $P < 0.05$ means that the intensity or the directionality of α-SMA$^+$ signals are statistically higher than those of VE-cad$^+$ signals. Four independent mouse brains (female, 6-15 months) were used for analysis (see Supplementary Fig. 4). Source data are provided as a Source Data file. **c** The 3D images of the original images with α-SMA$^+$ and VE-cad$^+$ signals in the brain. The representative 3D whole-brain images (left, sagittal view, YZ), the enlarged 3D images in the midbrain and pons areas (middle), and in the surface of isocortex area (right) are shown.

features of the signals (Fig. 6d). The Sliced Wasserstein kernel was used to evaluate the geometric features of the lymphatic vessels and extract the information about the pairwise distances among a set of lymphatic vessel objects, and their features were represented by a two-dimensional persistent diagram (Fig. 6d, Supplementary Fig. 9). In addition, the lymphatic vessel objects were coordinated in a two-dimensional plot that displays the relative positions of the vessel objects from the distances between them using MDS (Fig. 6e). The results showed that the lymphatic vessels in the fibrotic lung had specific structural features compared to those in the saline-treated or untreated lungs (Fig. 6e). Our data suggested that PH is useful for evaluation of vascular structures by extracting geometric features.

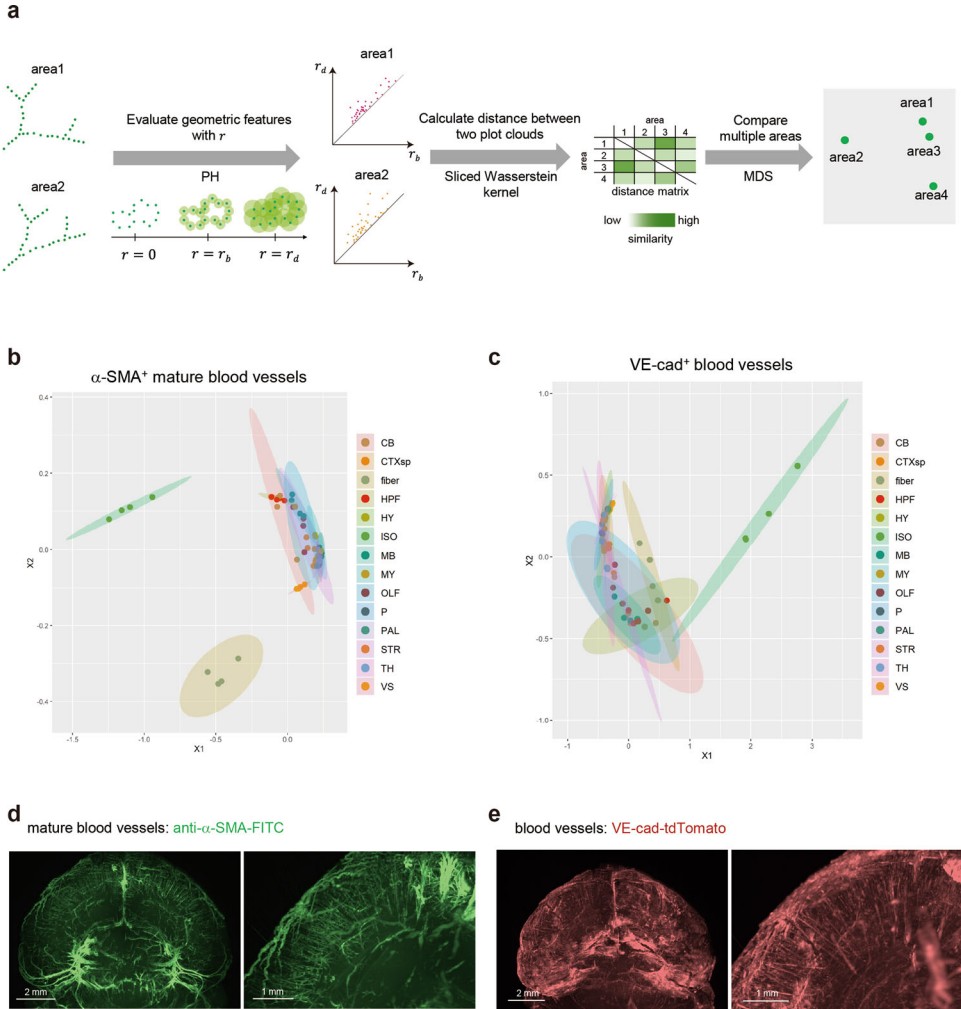

**Fig. 5 | Analysis of brain blood vessels using TDA. a** To extract the geometric features, persistent homology (PH), a main method of topological data analysis (TDA), is used. This method can evaluate structures by making virtual circles from the classified signals as a starting point. When the size of concentric circles is increased little by little, the larger circle will appear which connects them (birth time of circle). When the size of concentric circles become even larger, the appeared circles will disappear (dead time of circle). The radius of concentric circles at the birth time is defined as $r = r_b$, and the radius at the dead time is defined as $r = r_d$. These values ($r_b$ and $r_d$) are plotted as persistent diagram. To compare two areas based on their features, the distance between the two point clouds shown in PD is calculated by the Sliced Wasserstein kernel. Using the distance matrix obtained by calculating the distances between the point clouds in all pairs of areas by the Sliced Wasserstein kernel as an input, the proximity between areas from a geometric point of view is visualized by multidimensional scaling (MDS). **b, c** MDS plots showing blood vessels in the brain. The features of α-SMA⁺ mature blood vessels (**b**) or VE-cad⁺ blood capillaries (**c**) in each brain area were extracted with PH (14 brain areas: cerebellum (CB), cortical subplate (CTXsp), fiber tracts (fiber), hippocampal formation (HPF), hypothalamus (HY), isocortex (ISO), midbrain (MB), medulla (MY), olfactory areas (OLF), pons (P), pallidum (PAL), striatum (STR), thalamus (TH), and ventricular systems (VS)). The geometric features of α-SMA⁺ mature blood vessels or VE-cad⁺ blood capillaries in each area are shown in MDS. Four independent mouse brains (female, 6–15 months) were used for analysis. Source data are provided as a Source Data file. **d, e** The 3D brain images with α-SMA⁺ signals (**d**) and VE-cad⁺ signals (**e**). The representative 3D images of the isocortex area, showing the featured vascular structures inside the brain, are shown.

## Cancer metastasis and structure of lymphatic vessels

Next, we investigated the relationship between vascular structures and cancers. Lymphatic vessels play an important role not only in normal development and inflammation, but also in cancer metastasis. Therefore, we decided to investigate the positional relationship between cancer cells and lymphatic capillaries using an experimental lung metastasis model. Mouse melanoma B16F10 cells expressing mCherry were inoculated intravenously (i.v.), and the position of lymphatic vessel and cancer cells were monitored at 24 h and at 4 and 10 days after inoculation (Fig. 7a, b). The signals were classified as cancer colonies or lymphatic vessels using ilastik (Fig. 7c). The autofluorescence signals from trachea or bronchi were so high to mask the signals from B16F10 cells in the original images (Fig. 7c). Then, we studied the distance between B16F10 tumors and lymphatic vessels based on the pixel

classified signals. The 2D images confirmed the location of tumor colonies and lymphatic vessels (Fig. 7d). At 24 h after inoculation, most cancer cells and lymphatic vessels, particularly GFP⁺ lymphatic endothelial cells, did not co-exist (Fig. 7d, e, Supplementary Fig. 10a) and this tendency persisted until day 4 (Fig. 7e, Supplementary Fig. 10a). However, at day 10, most tumor colonies co-existed with lymphatic vessels (Fig. 7e, Supplementary Fig. 10a). From these data, we can speculate that metastases in the lymphatic node might be induced from these co-existing sites. Total signals of lymphatic vessels after the classification indicated that there was no big difference between the days after inoculation (Fig. 7f). However, the lengths and radii of branches were smaller at day 4 compared to those in the control lung (Fig. 7f). The classified signals as lymphatic vessels were also analyzed with TDA to extract the geometric features and represented by a two-

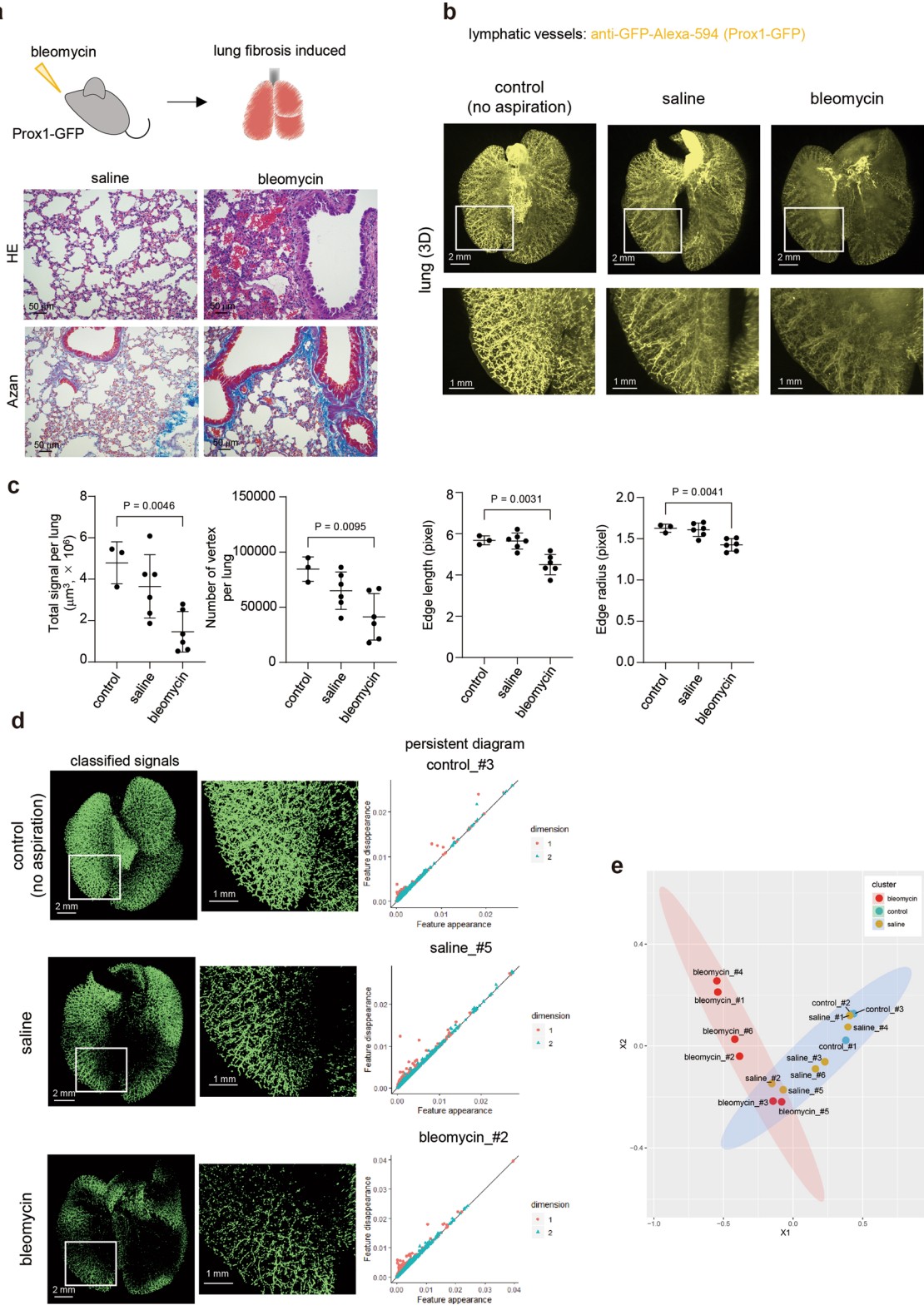

dimensional persistent diagram (Fig. 7g, Supplementary Fig. 10b). The lymphatic vessel objects were plotted by a two-dimensional plot reflecting the relative differences between the vessel objects evaluated by the Sliced Wasserstein kernel (Fig. 7g). The results revealed that lymphatic vessels in the B16F10 tumor-bearing lung were different from those in the normal lung, especially at day 4 (Fig. 7g). These results indicate that CUBIC not only enables quantification of metastatic tumors, but also spatiotemporal analysis of the structure of lymphatic vessels by PH.

## Discussion

Homeostasis of animal body is sustained by two branched networks, i.e., the blood and lymphatic vasculatures. However, these structures have not been fully examined at the whole organ level due to the lack of appropriate 3D imaging systems. In the present study, we successfully visualized mouse blood and lymphatic vessels in various organs using CUBIC 3D tissue-clearing imaging. Moreover, using these 3D images, we established a quantification method in combination with mathematical frameworks. To our best knowledge, this is the first

**Fig. 6 | Analysis of lymphatic vessels in the pulmonary fibrosis model using TDA. a** Induction of pulmonary fibrosis using bleomycin. Mice were instilled with bleomycin intratracheally (1.25–2.5 mg/kg) in Prox1-GFP mice (male, 4–16 months). Saline was instilled in the control group. The lungs were subjected to hematoxylin-eosin (HE) and Azan staining. Representative images from three independent experiments are shown. **b** The 3D whole-lung images of lymphatic vessels in the fibrosis model. Prox1-GFP mice were instilled intratracheally with bleomycin or saline. The mice were then sacrificed, and the lungs were subjected to CUBIC procedures. Representative 3D whole-lung images of lymphatic vessels are shown (Z = 10 μm step, digital zoom: 1.25). The enlarged 3D images (white insets) are shown in the bottom. **c** Quantification of lymphatic vessels (volume, branching/end points, and lengths and radii of branches). All the classified signals as lymphatic vessels are counted as a volume at the whole lung level. Using TubeMap, the number of branching/end points (vertex), and lengths and radii of branches were calculated. Data are shown with mean ± SD. Representative data from two independent

experiments are shown (control (no aspiration); *n* = 3 mice, saline; *n* = 6 mice, bleomycin; *n* = 6 mice). One-way analysis of variance (ANOVA) and Dunnett's multiple comparisons test were used. Source data are provided as a Source Data file. **d** The classified signals as lymphatic vessels in the lung fibrosis model. Classified signals as lymphatic vessels after training with ilastik software are shown (left). Persistent diagram of mouse lung lymphatic vessels in the fibrosis model. PH was applied to extracted signals. Persistent diagrams are shown in each sample (right). The red points (dimention1) and blue triangles (dimention2) represent planar feature points (loop) and spatial feature points (void), respectively, observed by the persistent homology method. Source data are provided as a Source Data file. **e** The geometric features of lymphatic vessels in the lung fibrosis model. To calculate the distance of each features in these persistent diagrams, the Sliced Wasserstein kernel is applied[81]. The results of (**d**) are shown in MDS. Representative data from two independent experiments are shown (control (no aspiration); *n* = 3 mice, saline; *n* = 6 mice, bleomycin; *n* = 6 mice). Source data are provided as a Source Data file.

study to conduct a structural analysis of mouse vasculatures using a combination of tissue-clearing technology and TDA. We believe that this approach will become a useful modality for the structural analysis of vasculatures.

Our results are in line with those of many previous studies and also demonstrate insights in networks of blood/lymphatic vessels in mouse organs, including the brain, kidney, lung, and intestine[16,34,40,49–51]. For example, many researchers have reported that lymphatic capillaries exist in the center of each villus in the intestine, surrounded by blood capillaries[52], and those findings were clearly visualized with 3D images in this study. Further, the lung lymphatic vessels seen here with 3D images were consistent with those in a previous study that used 2D IHC with CD90 staining[50,53]. Regarding brain lymphatic vessels, we showed that they were running along with the superior sagittal sinus, transverse sinus, and ventricle, according to the previous report[54]. In the kidney, we found that classified signals of lymphatic vessels seem to be running in parallel with the α-SMA⁺ artery or sympathetic nerves[34]. The network of lymphatic vessels in the mouse heart can also be seen from the dorsal or ventral side[55,56]. Regarding stomach, it is interesting that the blood capillaries and lymphatic vessels did not go along with each other. Recent reports also showed that tissue-clearing technology is useful for imaging of capillaries[16,51,56–61]. We thus believe that tissue-clearing technology will be useful for studying capillaries from a new perspective.

In the present study, we mainly used transgenic mice, but antibody immunostaining such as with anti-VEGFR3 antibody is also applicable to tissue-clearing imaging. Recent studies have reported advanced 3D staining protocols that will be helpful for further investigations on the various types of anatomical structures[17,58,62]. In addition, tomato-lectin or dextran injection is also useful for the visualization of blood vessels[16]. In our current study, α-SMA, which is expressed in the pericytes covering the blood vessels, is used as a marker of mature blood vessels in mouse brain and VE-cadherin as a marker of blood capillaries. The result from NHPP showed that the mature blood vessels existed in the midbrain to brainstem, which was consistent with the previous inspections[47,48].

Machine learning using ilastik software also successfully classified and extracted target signals from original images at pixel base to reduce autofluorescence signals and detect target signals, which could be masked by autofluorescence signals as previously described[44]. Recently, some studies describing signal segmentation of vasculatures utilizing machine learning have been reported[63,64]. Ilastik software is a helpful segmentation tool and is used for counting cell numbers or monitoring the amounts extracellular matrix (ECM) such as elastin[42,43,65]. Currently, measuring volumes or counting the dividing points remains the primary method for quantification of vasculatures. However, these approaches are inadequate for evaluation of the structural difference and do not take advantage of 3D imaging. We aimed to visualize the vascular networks at the whole organ level and to establish an evaluation method for 3D structure images. Therefore,

to extract morphological features from 3D images, we applied PH, a TDA method, and successfully assessed the difference in structures of lymphatic vessels. For quantification of structural differences of vasculatures, evaluation of volumes, widths, branching points, or the lengths between branching points based on 2D images is still the most common way. These parameters provide us critical information of vascular structures; however, some spatial information is lacking. If we try to get spatial information from 2D images, it could be a time-consuming and labor-intensive work (Supplementary Table 1). Current reports introduced some brand-new methods using various parameters including branching points and tortuosity from 3D images (Supplementary Table 1). We also evaluated our data with these parameters, such as the number of branching points or branch radii, and the results were consistent with our TDA or NHPP data. Compared to other existing quantification methods, PH is not focusing on one specific parameter, but it enables us to understand their structures more inclusively using virtual circles. Therefore, this method has a potential to detect structural differences from 3D images, which cannot be detected by other existing methods. NHPP is a quantitative parameter based on intensity, which can also evaluate directionalities.

Meanwhile, our methods also have some limitations. First, the quality of original images can affect the result of NHPP and PH, which can be improved in the future through the use of more superior microscopes and improved classification processes. Although ranges of proper thresholds that can detect the structural differences using PH are not narrow (Supplementary Fig. 7), the manual classification process should be automated to reduce the bias. It should also be noted that these drawbacks can be seen in other existing methods. In addition, for running our current method, some procedures require a computer with high specifications, which could limit their applications (Supplementary Table 2). Thus, although our current methods need to be improved in some points, we believe that this is a valuable study to use mathematical frameworks such as PH and NHPP in 3D capillary images captured by tissue-clearing technologies (Supplementary Fig. 11).

To understand the molecular mechanisms of cancer metastasis, it is important to visualize not only cancer cells, but also other stromal cells in the tumor microenvironment including vasculatures. A previous report showed that condensed blood vessels existed inside the bones, where cancer metastasis are often observed[66]. Regarding B16F10 cancer metastasis, B16F10 cancer cells did not co-exist with lymphatic vessels at 24 h after i.v. inoculation, with most tumor colonies interacting with lymphatic vessels on day 10.

Lymphangiogenesis and lymphatic vessel remodeling play an important role in cancer metastasis[67]. In melanoma patients, remodeling of the lymphatic capillaries is related to poor prognosis[68]. Our B16F10 metastasis model showed that B16F10 cancer cells can utilize the pre-existing lymphatic vessels, because they are located close to tumor colonies during the early stage of metastasis. However, we also

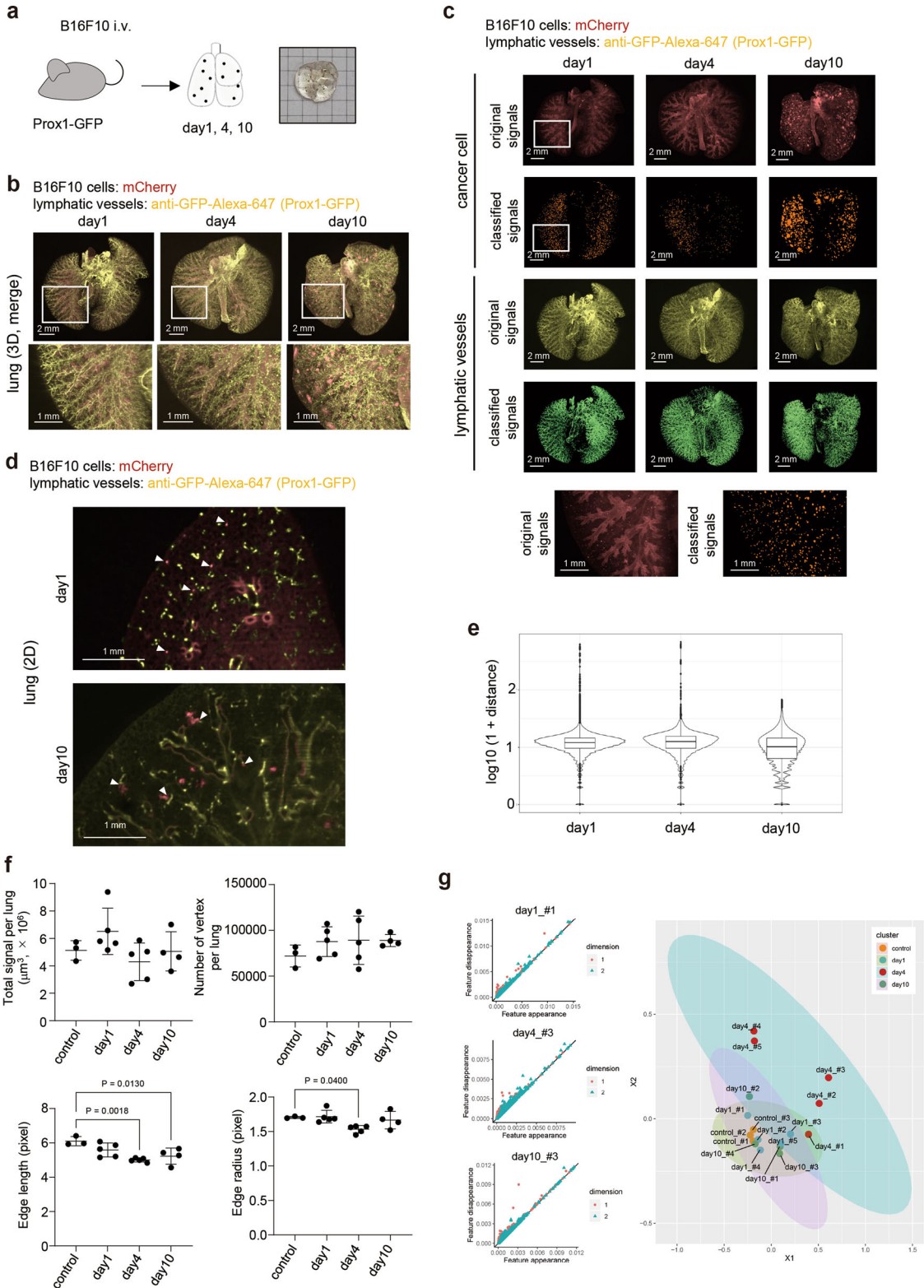

speculated that lymphatic capillaries change both physically and qualitatively during the proliferation of B16F10 cells. Further studies to validate this hypothesis should include staining of specific markers of developing lymphatic vessels, not pre-existing ones. Our data imply that the structures have altered due to the inflammation induced by cancer cell inoculation, which might result in vascular leakage. Although imaging of the interaction between lymphatic vessels and cancer cells tends to be conducted in specific local areas because of the limitation of imaging field, it is also important to detect the

changes of lymphatic vessels at the whole organ level. Our system is thus a promising way to capture the architectural changes during various pathological conditions, including cancer and fibrosis at the whole organ level.

In conclusion, our system is a promising method to capture the structural changes of the vasculature at the whole organ level. Our results support that this method is a useful approach for 3D structure analysis and is helpful for investigating the underlying mechanisms of various diseases.

**Fig. 7 | Analysis of lymphatic vessels in B16F10 lung metastasis model. a** Scheme of the experimental procedures. The lung image with B16F10 metastatic colonies after CUBIC procedures is shown. **b** The 3D whole-lung images at days 1, 4, and 10 after intravenous B16F10 injection. B16F10 cells expressing mCherry were inoculated intravenously in Prox1-GFP mice (female, 3–5 months). At 1, 4, and 10 days after inoculation, the excised lungs were subjected to CUBIC procedures. Representative 3D whole-lung images of lymphatic vessels (yellow) and cancer cells (red) are shown. The enlarged 3D images (white insets) are shown in the bottom ($Z$ = 10 μm, digital zoom:1.25). **c** The 3D whole-lung images of original signals and classified signals using ilastik software from B16F10 cancer metastasis are shown. The enlarged 3D images (white insets) are shown in the bottom. **d** The 2D images at days 1 and 10 after intravenous B16F10 injection. Representative 2D (XY) images are shown. White arrowheads indicate cancer cells and tumor colonies. **e** The distances between B16F10 tumors and lymphatic vessels. The distances between B16F10 cancer cells or tumors and lymphatic vessels (GFP⁺ lymphatic endothelial cells) at days 1, 4, and 10 were calculated using the 3D images (pixel-based, $n$ = 3 mice per group). Representative data from 2 independent experiments are shown. These boxplots show five-number summary statistics; i.e., the middle thick line, two hinges, and two whisker correspond to the median, interquartile range (IQR), and

the largest and the smallest values unless "outliers", respectively. "Outliers" are values which are larger or smaller than from 1.5 IQR, and shown in the black points. **f** Quantification of lymphatic vessels (volume, branching/end points, and lengths and radii of branches). All the classified signals as lymphatic vessels are counted as a volume at the whole lung level. With all the classified signals as lymphatic vessels, branching/end points (vertex), and lengths and radii of branches were calculated using TubeMap. The results from 2 independent experiments are shown (control; $n$ = 3 mice, day 1; $n$ = 5 mice, day 4; $n$ = 5 mice, day 10; n = 4 mice). Data are shown with mean ± SD. One-way analysis of variance (ANOVA) and Dunnett's multiple comparisons test were used. Source data are provided as a Source Data file. **g** The geometric features of lymphatic vessels in B16F10 experimental metastasis model using PH. The extracted signals of the lymphatic vessels were subjected to PH. Persistent diagrams are shown (left). The red points (dimention1) and blue triangles (dimention2) represent planar feature points (loop) and spatial feature points (void), respectively, observed by the persistent homology method. The results from 2 independent experiments are also shown in MDS (control; $n$ = 3 mice, day 1; $n$ = 5 mice, day 4; $n$ = 5 mice, day 10; $n$ = 4 mice). Source data are provided as a Source Data file.

## Methods

### Ethical statement
All experiments were approved by and carried out according to the guidelines of the Animal Care and the Use of Committee of the Graduate School of Medicine, The University of Tokyo.

### Mice
C57BL/6J mice (5w, female) for lectin injection were purchased from Sankyo Lab Service (Japan). Cdh5-BAC-Cre^ERT2 mice were generated in the previous study[69]. ROSA-lox-stop-lox-tdTomato mice were purchased from Jackson Laboratory, and Prox1-GFP mice were purchased from Mutant Mouse Resource Research Centers (MMRRC). The background of all strains is C57BL/6J. We generated Cdh5-BAC-Cre^ERT2; ROSA-lox-stop-lox-tdTomato (VE-cad-tdTomato) mice by crossing Cdh5-BAC-Cre^ERT2 mice with ROSA-lox-stop-lox-tdTomato mice. In addition, we also generated Prox1-GFP; VE-cad-tdTomato mice by crossing Prox1-GFP mice with VE-cad-tdTomato mice. To express Cre recombinase, Tx (#T5648, Sigma-Aldrich) dissolved in oil (20 mg/ml) was inoculated intraperitoneally (i.p.) (112.5 mg/kg) for 4–5 days. Mice were maintained in 12 h/12 h dark/light cycle at 20–23 degree with 40–60% humidity.

### Reagents
Lycopersicon esculentum lectin (tomato-lectin), FITC, and Texas-Red conjugate (#FL-1171, #TL-1176) were purchased from VECTOR Laboratory. Anti-α-SMA-FITC (#F3777, Sigma-Aldrich), anti-GFP antibody conjugated with Alexa Fluor 594 (#A21312) or Alexa Fluor 647 (#A31852), and anti-goat 2nd antibody conjugated with Alexa Fluor 647 (#A21447) were purchased from ThermoFisher Scientific. Anti-VEGFR3 antibody (#AF743) was purchased from R&D systems.

### CUBIC procedure
Tissue clearing was performed using the CUBIC regents[10,12,13]. Mice were sacrificed and perfused with 20–30 ml phosphate buffered saline (PBS) and 20–30 ml 4% PFA (Wako) in PBS. The organs were post-fixed with 4% PFA at 4 °C overnight. Then, samples were washed with PBS for more than 6 h with three times exchange to remove PFA. After washing, samples were immersed in 50% CUBIC-L, which is a mixture of 10 w% polyethylene glycol mono-*p*-isooctylphenyl ether (Triton-X, Nacalai Tesque) and 10 w% *N*-butyldiethanolamine (Tokyo Chemical Industry), at 37 °C for over 6 h and further immersed in 100% CUBIC-L at 37 °C for 1–5 days (Supplementary Table 3), depending on the organ. After decolorization and delipidation with CUBIC-L, the samples were washed with PBS for more than 6 h with three times refreshing. Then, the samples were immersed in 50% CUBIC-R (N), which is a mixture of

45 w% 2, 3-dimethyl-1-phenyl-5-pyrazolone (antipyrine, Tokyo Chemical Industry) and 30 w% nicotinamide (Tokyo Chemical Industry), for over 6 h and further immersed in 100% CUBIC-R (N) at room temperature to adjust the RI. For antibody staining, organs were immersed in antibody dilution (Supplementary Table 4) with staining buffer (1% casein, 0.01% NaN₃, 0.5% Triton-X). All procedures were performed with gentle shaking. For samples that were thin or easily moved during microscopy (e.g., the skin), they were embedded in 2% gel after delipidation and decolorization with washing. Then, the embedded samples were immersed in CUBIC-R (N) similar to that for non-embedded samples.

To enhance the signals from GFP, samples from Prox1-GFP mice were stained with anti-GFP antibody for 3-4 days. Anti-α-SMA antibody labeled with FITC was used to visualize mature blood vessels. For tomato-lectin staining of blood vessels, tomato-lectin conjugated with Texas-Red or FITC was injected intravenously. The mice were then sacrificed and perfused with PBS and PFA at 5 min after lectin injection.

### Microscopy
Whole-body and whole organ images were captured using a custom-built LSFM developed by Olympus, which were used in our previous study[12]. The wavelengths of excitation laser were 488, 532, 590, and 639 nm and (488 nm: FITC, 532 nm: tdTomato, 590 nm: mCherry, Texas-Red, and Alexa Fluor 594, 639 nm: Alexa Fluor 647). The emission filters were 495-540 nm (Φ32 mm, FITC), 610-640 nm (Φ 32 mm, mCherry, TexasRed, Alexa Fluor 594), 660-750 nm (Φ32 mm, Alexa Fluor 647) and 590-650 nm (Φ25 mm, tdTomato). During acquisition, samples were immersed in a mixture of HIVAC-F4 (Shin-Etsu Chemical) and mineral oil (#M4810, Sigma-Aldrich). Regarding custom-made LSFM, images were captured at 0.63 × objective lens (numerical aperture = 0.15, working distance = 87 mm) with digital zoom from 0.8 × to 6.3 × zoom. This LSFM is equipped with sCMOS camera (NEO5.5, ANDOR) and their pixel number is 2560 × 2160. Therefore, the pixel resolution is as follows; digital zoom 0.8 ×; pixel resolution ($x$) and ($y$): 12.9 μm, digital zoom 1 ×; pixel resolution ($x$) and ($y$): 10.32 μm, digital zoom 1.25 ×; pixel resolution ($x$) and ($y$): 8.25 μm, digital zoom 1.6 ×; pixel resolution ($x$) and ($y$): 6.45 μm, digital zoom 2 ×; pixel resolution ($x$) and ($y$): 5.16 μm, digital zoom 2.5 ×; pixel resolution ($x$) and ($y$): 4.13 μm, digital zoom 4.0 ×; pixel resolution ($x$) and ($y$): 2.58 μm, digital zoom 6.3 ×; pixel resolution ($x$) and ($y$): 1.64 μm. Images were captured with $z$ = 10 μm step. Each 2D image taken by left and right laser was merged with max intensity and used as one plane image. All raw image data were collected in a lossless 16-bit TIFF format. To show the images with original signals, the tiff images captured by LSFM were converted using the Imaris File Converter and analyzed by the Imaris software (version 8.4) and Free Imaris Viewer (version 9.5).

## Pulmonary fibrosis induced by bleomycin intratracheal instillation

Pulmonary fibrosis was induced by bleomycin intratracheal instillation[70]. Bleomycin sulfate (#9041-93-4, Toronto Research Chemicals) was dissolved in saline solution (Otsuka) (1 mg/ml). Mice were anesthetized, and the dissolved bleomycin was administered intratracheally via oropharyngeal aspiration (1.2–2.5 mg/kg) (Prox1-GFP mice, male, 4–16 months). At 10 to 14 days after instillation, the mice were sacrificed, and the lungs were subjected to CUBIC procedures or 2D IHC.

## Experimental lung metastasis models

Mouse melanoma B16F10 cells (American Type Culture Collection) were maintained in Dulbecco's Modified Eagle's Medium containing 10% fetal bovine serum (FBS, Gibco), penicillin, and streptomycin. A stable B16F10 transfectant expressing mCherry (B16F10-mCherry) was established by infection of lentiviral vectors[71]. B16F10-mCherry cells were inoculated intravenously in Prox1-GFP mice (female, 3–5 months) ($3 \times 10^5$ cells/500 μl/mouse). On days 1, 4, and 10 after inoculation, mice were sacrificed and perfused with 4% PFA, followed by CUBIC analysis of the lungs.

## Histological analysis

After perfusion with PFA, the samples were subjected to sucrose replacement from 5% up to 20%. After sucrose exchange, samples were frozen with O.C.T. compound (Sakura Finetek, Japan). To prepare the paraffin sections, the excised organs were fixed in 4% PFA and embedded in paraffin. The samples were then subjected to hematoxylin and eosin staining and Azan staining. For IHC, frozen sections were subjected to immunostaining with anti-LYVE1 antibody (#ab14917, Abcam) and anti-CD31 antibody (#553370, BD Pharmingen). For nucleus staining, slides were encapsulated with mounting compound containing DAPI (#0100-20, Southern Biotech).

## Pixel classification by ilastik

For pixel classification, a machine-learning algorithm was used[44,46]. Before the classification, tiff images captured by LSFM were converted to hdf5 files using python (https://www.python.org/). Then, these hdf5 files were trained using ilastik software (https://www.ilastik.org/)[65], which is an interactive machine learning-based image analysis software[72]. This software uses a random forest classifier in the learning step, where neighborhood of each pixel is characterized by a set of generic (nonlinear) features[65]. Hyperparameters related to this random forest algorithm were automatically configured by ilastik. In the training phase, fine-tuning can be done by interactively providing new labels. In this study, 2 annotations were defined as "target signal" or "background signal" for the training of pixel classification. Following the ilastik workflow, the descriptors were selected as follows; Gaussian Smoothing of Intensity from 0.3 px to 10 px, Laplacian of Gaussian of Edge from 0.7 px to 10 px, Gaussian Gradient Magnitude of Edge from 0.7 px to 10 px, Difference of Gaussians of Edge from 0.7 px to 10 px, Structure Tensor Eigenvalues of Texture from 0.7 px to 10 px, and Hessian of Gaussian Eigenvalues of Texture from 0.7 px to 10 px. The process of annotating the pixels, evaluating the prediction map and re-annotating the pixels was repeated until the prediction map is correct. To increase the robustness several samples were annotated with identical labeling conditions. Then, the probabilities of "target signal" intensities were exported as an h5 file. When the images of probabilities of classified target signals were shown, Imaris software was used. The threshold of these probabilities was decided by comparing the original signals and pixel classified signals using Imaris software side by side. To avoid the bias by this manual threshold, data were checked by two researchers independently. Data processing was performed in python using custom scripts based on standard packages including Scipy, Numpy, Pandas and h5py as described before[44]. We used the python code shown in our previous study (https://gitlab.com/TGFbeta/kubota_tgfb)[44]. In case of further analysis with TDA, NHPP and TubeMap, classified signals were used.

## Anatomical annotation in the brain with CUBIC-Atlas

For anatomical annotation in the brain, Advanced Normalization Tools (ANTs) were used as a registration tool[73]. Using ANTs, affine transformation and then nonlinear transformation were performed as previously described[32,46,73]. Through registration with ANTs, original mouse brain images can be transformed and compatible to registration data. Then, anatomical brain information in CUBIC-Atlas (version1.1) (http://cubic-atlas.riken.jp/), which were based on Allen Brain Atlas (version3) (https://portal.brain-map.org/), were added to these brain transformed data[45,46]. Fourteen brain areas are as follows; cerebellum (CB), cortical subplate (CTXsp), fiber tracts (fiber), hippocampal formation (HPF), hypothalamus (HY), isocortex (ISO), midbrain (MB), medulla (MY), olfactory areas (OLF), pons (P), pallidum (PAL), striatum (STR), thalamus (TH), and ventricular systems (VS).

## Topological data analysis (TDA) and persistent homology (PH)

Our approach was based on TDA, which quantifies geometric features and patterns of vessel structures. TDA is a new technique that extracts useful shape features from large and complex data sets, and it has already been applied in various biological contexts[74], including studies of gene expression at the single-cell level[75], viral reassortment[25], horizontal evolution[76], cancer genomics[26], and other complex diseases[77,78]. We used PH, which is one of the main methods in TDA and is developed from computational topology and algebraic topology.

TDA is an algebraic method to identify the topological features of data, which include connected components (or clusters), loops, and holes. TDA approximates a continuous geometry by building a simplicial complex that is an object built from points, edges, triangular faces, and other topological features of the given data. A point is a zero-dimensional simplex, an edge between two points is a one-dimensional simplex, a triangular face is a two-dimensional simplex, and a solid tetrahedron is a three-dimensional simplex, and so forth for higher dimensional simplexes. Connecting several simplexes with the intersection of any two simplexes results in a simplicial complex. An example of simplicial complexes is illustrated below.

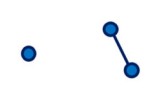 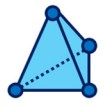 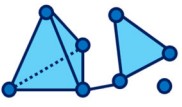

0-simplex  1-simplex  2-simplex  3-simplex  simplicial complex

Given a set of $n$ data points $S = \{x_i\}_{i=1}^{n}, x_i \in R^p$, we started building simplicial complex representations of $S$ by replacing a ball of radius $r$ centered at each point $x_i \in S$ to obtain $\{B_r(x_i) = \{y \in D : d(x_i, y) \leq r\}\}_{i \in I}$. The simplicial complex representation of $S$ with respect to $r$ is given by the union of all $k$-dimensional simplices such that $B_r(x_{jl}) \cap B_r(x_{jm}) \neq \varnothing$ for all $l, m = 0, 1, \ldots, k$. This is known as the Veitoris-Rips complex of $S$ with respect to $r$. In the analysis of simplicial complexes, their topological properties are based on the homology theory, which defines the topological properties of any given dimension of a space. These properties can be represented by homology groups $H_0$, $H_1$, $H_2$, and other higher groups. In particular, the zero-dimensional homology group $H_0$ represents the connected components of Veitoris-Rips complex, the one-dimensional homology group $H_1$ represents one-dimensional loops, and the two-dimensional homology group $H_2$ represents two-dimensional voids. For more technical details on homology, see Refs. 79,80.

Inevitably, the homology group of simple complex representations of S is sensitive to the choice of the scaling parameter $r$. To overcome this problem, PH tracks the changes in the homology group according to the different scaling parameters, as illustrated in Fig. 5a.

Each point is associated with a sphere of the same size and the radius of the sphere is used as a scaling parameter $r$ (Fig. 5a). By expanding the radius of each point $r$, homology groups are constructed for different values of scaling parameters.

The results from PH can be represented as pairs of birth time and death time of the topological feature. More specifically, for each topological feature, the birth time ($r_b$) and death time ($r_d$) represent the scaling parameter $r$ at which the topological feature is born and vanished, respectively. The persistence of a topological feature is given by its lifetime $r_d - r_b$.

To visualize global geometric features and patterns of vessel structures from PH, we used a persistence diagram where each point corresponded to a topological feature, and the $x$- and $y$-coordinates of the points were the birth and death time, $r_b$ and $r_d$, respectively. In a persistence diagram, a topological feature that persists across a broader range of parameter $r$ lies farthest away from the diagonal line $y = x$. This visualization provides a low-dimensional and descriptive representation of the data $S$ and is therefore useful for a range of data-driven tasks.

## Comparison of multiple persistence diagrams

To extract robust topological features from multiple PDs and compare them, we used a Sliced Wasserstein kernel for PDs that quantified the difference between two persistence diagrams based on Sliced Wasserstein metric[81]. Given $\boldsymbol{\theta} \in \mathbb{R}^2$ with $\|\boldsymbol{\theta}\|_2 = 1$, let $L(\boldsymbol{\theta})$ denote the line through zero $\{\lambda\boldsymbol{\theta}, |\lambda \in \mathbb{R}\}$, and $\pi_{\boldsymbol{\theta}} : \mathbb{R}^2 \to L(\boldsymbol{\theta})$ be the orthogonal projection onto $L(\boldsymbol{\theta})$. Let $\mathrm{Dg}_i$ and $\mathrm{Dg}_j$ be two PDs and let

$$\mu_1^{\boldsymbol{\theta}} = \sum_{p \in \mathrm{Dg}_1} \delta_{\pi_{\theta}(p)}, \mu_{1\Delta}^{\boldsymbol{\theta}} = \sum_{p \in \mathrm{Dg}_1} \delta_{\pi_{\theta} \circ \pi_{\Delta}(p)}, \mu_2^{\boldsymbol{\theta}} = \sum_{p \in \mathrm{Dg}_2} \delta_{\pi_{\theta}(p)}, \text{ and }$$

$$\mu_{2\Delta}^{\boldsymbol{\theta}} = \sum_{p \in \mathrm{Dg}_2} \delta_{\pi_{\theta} \circ \pi_{\Delta}(p)}$$

where $\pi_{\Delta}$ is the orthogonal projection on the diagonal. Then, the Sliced Wasserstein distance is defined by:

$$\mathrm{SW}(\mathrm{Dg}_i, \mathrm{Dg}_i) \stackrel{\text{def}}{=} \frac{1}{\pi} \int_{-\frac{\pi}{2}}^{\frac{\pi}{2}} W\left(\mu_1^{\theta} + \mu_{2\Delta}^{\theta}, \mu_2^{\theta} + \mu_{1\Delta}^{\theta}\right) \mathrm{d}\theta$$

where $W(\mu, v)$ is the 1-Wasserstein distance given by:

$$W(\mu, \nu) = \inf_{P \in \Pi(\mu, \nu)} \iint_{\mathbb{R} \times \mathbb{R}} |x - y| P(\mathrm{d}x, \mathrm{d}y).$$

For a scale parameter $\sigma > 0$, the Sliced Wasserstein kernel between $\mathrm{Dg}_i$ and $\mathrm{Dg}_j$ is defined as:

$$k_{\mathrm{SW}}\left(\mathrm{Dg}_i, \mathrm{Dg}_j\right) \stackrel{\text{def}}{=} \exp\left(-\frac{\mathrm{SW}(\mathrm{Dg}_i, \mathrm{Dg}_i)}{2\sigma^2}\right).$$

The Sliced Wasserstein kernel is the total distance between matched loop structures and gives an overall quantification of the global similarity between two PDs.

To visualize and compare multiple PDs, the PDs were embedded into lower dimensional space. Given the Sliced Wasserstein kernel matrix $K = \left[k_{\mathrm{SW}}\left(\mathrm{Dg}_i, \mathrm{Dg}_j\right)\right]_{1 \le i, j \le n}$ for all PDs, we used the classical MDS, which is a two-dimensional reduction method[82]. In our analysis, MDS maps the information about the geometric distances between individuals using the Sliced Wasserstein kernel, into an abstract Cartesian space. Persistent diagrams based on persistent homology and the Sliced Wasserstein distances were calculated using HomCloud (version 3.0.1) and the R package kernel TDA (version 1.0.0), respectively.

## Non-homogeneous Poisson process (NHPP)

We considered blood vessels and lymphatic vessels as points in three-dimensional space and fitted them to an NHPP model. NHPP is characterized by the intensity function of the points. In this study, intensity is the mean of number of the points in the unit space. We considered a semiparametric intensity function using a kernel function and used the following logistic kernel:

$$g(u) = \prod_{i=1}^{3} a \, b_i / (e^{b_i u_i} + 2 + e^{-b_i u_i})$$

where index 1, 2, and 3 correspond to the $x$, $y$, and $z$ axes, respectively.

The intensity of any point $u = (u_1, u_2, u_3)$ is given by:

$$\lambda(u) = \sum_{u_j \neq u} g(u - u_j)$$

The likelihood function of NHPP was defined by the following equation[83]:

$$L(\theta) = \prod_j \lambda(x_j) \exp\left(-\int_0^T \lambda(u) \mathrm{d}u\right)$$

Here, $\theta = (a, b_1, b_2, b_3)$ were the unknown parameters to be estimated.

The parameter $a$ controlled the overall intensity. The parameters $b_i$ ($i = 1, 2, 3$) were the concentration parameter, and the higher the value, the more concentrated was the effect of the originating point around the point.

We compared α-SMA⁺ mature blood vessels and VE-cad⁺ blood capillaries using the estimated parameters as features. Fisher's exact test was performed as follows:

1. Extract the part where parameter $a$ has changed more than twice and where at least one reciprocal of $b_x$, $b_y$, and $b_z$ has changed more than twice.
2. Make a 2 × 2 table such as Table 1 and perform Fisher's exact test.

If the parts that changed were random, the ratio of the extracted parts did not change from the overall ratio. If the changing part was concentrated in a specific area, the ratio of that part will increase. Non-homogeneous Poisson process model was estimated with our original R functions. Fisher's exact test was performed and several plots were generated using R (version 3.6.2).

## Evaluation of vascular structures using "TubeMap"

To evaluate vascular structures with parameters such as branching points, we used a TubeMap package (https://github.com/ChristophKirst/ClearMap2), which is a pipeline for evaluation of vascular structures, especially in brain[20]. In a brain vasculature analysis, the images showing α-SMA⁺ vasculatures were used to register to Allen Brain Atlas with elastix package[84,85]. In this analysis, the images with classified signals by ilastik were used and constructed graph network with the vascular branch points as vertices. To construct graph network, signals were skeletonized with customized algorithms[86], and defined as branching points, connecting points, and end points. The radius of each point was calculated by measuring the distance to the nearest voxel whose intensity is less than half of the point. The skeletonized network was reduced to small graph with the branch points and end points as vertices. We segmented brain area into 14 regions (CB, CTXsp, fiber, HPF, HY, ISO, MB, MY, OLF, P, PAL, STR, TH, VS). Branching points and end points were defined as "vertex", and the parts between branching and/or end points were defined as "edge". The number of vertex and edge, the length of edge, and the radii of vertex and edge in each region were calculated. In the lung lymphatic vessel analysis, we constructed graph network without brain area registration, and calculated the number of vertex and edge, the length of edge, and the radii of vertex and edge.

**Table 1 | 2 × 2 table used in Fisher's exact test**

|                  | certain area | others |
|------------------|--------------|--------|
| Fold change >= 2 | A            | B      |
| No change < 2    | C            | D      |

## Statistics and reproducibility

For comparisons of the data, one-way analysis of variance (ANOVA) and Dunnett's multiple comparisons test, paired t-test and Bonferroni-Dunn method were used. Significant differences were defined as $P < 0.05$. Statistical analyses were performed with GraphPad Prism9 (GraphPad Software). No statistical method was used to predetermine sample size.

## Reporting summary

Further information on research design is available in the Nature Research Reporting Summary linked to this article.

## Data availability

Allen Brain Atlas and CUBIC-Atlas are available online (https://portal. brain-map.org/ and http://cubic-atlas.riken.jp/). The data for most figures are provided in the Source Data file. The data not provided in the Source Data file and representative raw image data (tiff files) are available upon request to the authors, due to the big data size, requests will be answered within 2 weeks. Source data are provided with this paper.

## Code availability

The analysis pipeline for this study is available at https://github.com/ nagoya-sysbiol/cubic_analysis. ANTs is available at http://stnava. github.io/ANTs/. The python code for TubeMap analysis can be accessed at https://github.com/ChristophKirst/ClearMap2.

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

## Acknowledgements

The authors thank Dr. Hiroyuki Miyoshi (deceased, formerly RIKEN) for providing plasmids, Dr. Tomoyuki Mano (Okinawa Institute of Science and Technology Graduate University) for technical assistance of machine learning, Dr. Etsuo A. Susaki (Juntendo University) for his expertise on the tissue-clearing method, Dr. Seiji Yamamoto (Toyama University) for his expertise on vascular biology, and all the members at Miyazono and Shimamura laboratories, especially Dr. Ryo Tanabe and Ms. Keiko Yuki for their supports. We also thank Bitplane for instruction in operating the Imaris software. This work was supported by grants from KAKENHI grants-in-aid for scientific research on Innovative Area on Integrated Analysis and Regulation of Cellular Diversity (K.M. and S.E., grant number 17H06326), KAKENHI grants-in-aid for scientific research (S) (K.M., grant number 15H05774; H.R.U., grant number JP25221004), KAKENHI grants-in-aid for scientific research (A) (K.M., grant number 20H00513), KAKENHI grants-in aid for scientific research (B) (T.S., grant number 20H04281), KAKENHI grants-in-aid for scientific research on Innovative Areas on Constructive Understanding of Multi-Scale Dynamism of Neuropsychiatric Disorders (T.S., grant number 19H05210), KAKENHI grants-in-aid for scientific research on Innovative Area on Transomics Analysis of Metabolic Adaptation (T.S., grant number 20H04841), KAKENHI grants-in-aid for Challenging Exploratory Research (T.S., grant number 20K21832), and KAKENHI grants-in-aid for early-career scientists (K.T., grant number 19K16604; S.I.K., grant number 20K16212; K.A., grant number 20K19921) from the Japan Society for the Promotion of Science (JSPS), RADDAR-J (T.S., grant number JP20ek0109488), Brain/MINDS Health and Diseases (T.S., grant number JP21wm0425007), Brain/MINDS (H.R.U., grant number JP21dm0207049), Science and Technology Platform Program for Advanced Biological Medicine (H.R.U., grant number JP21am0401011) from Japan Agency for Medical Research and Development (AMED), HFSP Research Grant Program (H.R.U., grant number HFSP RGP0019/2018) from Human Frontier Science Program (HFSP), grant-in-aid from Takeda Science Foundation (H.R.U.), Moonshot R&D (T.S., grant number JPMJMS2025), and ERATO (H.R.U., grant number JPMJER2001) from Japan Science and Technology Agency (JST).

## Author contributions

K.T. and K.M. designed the study. K.T., S.I.K., S.E., and Y.M. analyzed the 3D images. T.S., N.F., and K.A. conducted the mathematical analysis. Y.K., S.H., Y.Y., and T.W. maintained the transgenic mice. K.T., T.S., K.M., and H.R.U. wrote the manuscript. All authors discussed the results and commented on the manuscript text.

## Competing interests

The authors (K.M., S.E., and H.R.U.) declare the following competing interests. K.M. and S.E. were partly supported by Eisai, Co., Ltd. H.R.U. is a co-inventor on patent applications covering the CUBIC reagents (PCT/JP2014/070618 (pending), patent applicant: RIKEN, PCT/JP2017/016410 (pending), patent applicant: RIKEN) and CUBIC-HV reagents (PCT/JP2020/ 31840 (pending), patent applicant: CUBICStars) and a co-founder of CUBICStars. All other authors declare no competing interests. This work was partly done by technical support of Olympus Corporation.

## Additional information

[1]Department of Molecular Pathology, Graduate School of Medicine, The University of Tokyo, 7-3-1 Hongo, Bunkyo-ku, Tokyo 113-0033, Japan. [2]Laboratory of Medical Statistics, Pharmaceutical Science, Faculty of Pharmacy, Kobe Pharmaceutical University, 4-19-1 Motoyama-Kitamachi, Higashi-Nada-ku, Kobe, Hyogo 658-8558, Japan. [3]Division of Systems Biology, Graduate School of Medicine, Nagoya University, 65 Tsurumai-Cho, Showa-ku, Nagoya 466-8550, Japan. [4]Division of Pharmacology, Graduate School of Medical and Dental Sciences, Niigata University, 1-757 Asahi-machi-dori, Chuo-ku, Niigata 951-8510, Japan. [5]Institute for NanoSuit Research, Preeminent Medical Photonics Education & Research Center, Hamamatsu University School of Medicine, 1-20-1 Handayama, Higashi-ku, Hamamatsu, Shizuoka 431-3125, Japan. [6]Department of Anatomy, Keio University School of Medicine, 35 Shinanomachi, Shinjuku-ku, Tokyo 160-8582, Japan. [7]Department of Biochemistry, Graduate School of Medical and Dental Sciences, Tokyo Medical and Dental University (TMDU), 1-5-45 Yushima, Bunkyo-ku, Tokyo 113-8549, Japan. [8]Department of Systems Pharmacology, Graduate School of Medicine, The University of Tokyo, 7-3-1 Hongo, Bunkyo-ku, Tokyo 113-0033, Japan. [9]Laboratory for Synthetic Biology, RIKEN Center for Biosystems Dynamics Research, 1-3 Yamadaoka, Suita, Osaka 565-0871, Japan. [10]These authors contributed equally: Ko Abe, Shimpei I Kubota, Noriaki Fukatsu. ✉e-mail: shimamura@med.nagoya-u.ac.jp; miyazono@m.u-tokyo.ac.jp

