## [Peer Review File · Nature Communications]

Reviewers' Comments:

Reviewer #1:

Remarks to the Author:

The manuscript describes a new approach for the analysis of vascular structures in mouse brains and other organs. For imaging established light sheet microscopy and clearing technologies are used with good results. The new twist in this study is the combination with machine learning techniques. Using this combination the authors could substantiate existing hypotheses and findings that were based on older techniques like standard histology. As this manuscript adds a new item to the toolbox of 3D imaging of cleared organs and animals it can be expected that future studies will build on this advance and from my point of view the study can be published

Reviewer #2:

Remarks to the Author:

The manuscript 'A new analysis modality for vascular structures combining tissue-clearing technology and topological data analysis' reports three-dimensional (3D) imaging studies of the blood and lymphatic vasculature in different organs, using the CUBIC tissue clearing method in combination with mathematical frameworks. The purpose of the study is 'to evaluate the usefulness of the tissue-clearing technology CUBIC for visualizing blood and lymphatic vessels in the adult mouse.

The first part, corresponding to Figures 1-3, is dedicated to reproducing data already published by others in seminal papers on the CUBIC method, with the difference of using transgenic endothelial and lymphatic-specific reporter lines. The images shown are lacking definition and should be improved. The literature cited on the 3D-imaging of the blood and lymphatic vasculature is also incomplete. The conclusions drawn from these observations, such as 'these data support that CUBIC imaging is useful for visualization of blood vasculatures' or 'These data suggested organ- or area-specific structures of lymphatic vessels' may appear rather simplistic than providing new information on the organ-specific features of the vasculature.

The second part of the study aims at improving the quality of the vasculature analysis by using quantitative data extracted from the tiff images and processed by software and mathematical modeling. The procedure is applied to different organs, including the brain where it is used to cluster and distinguish arteries (SMA+) from other blood vessels (VE-cad+) based on parameters of intensity and geometric features. The procedure is only superficially explained, and the illustrating data (Figures 4C and 5C) are not convincing. It is hard to understand the benefit of this approach compared to other methods of quantifications, and the biological significance of the present clustering. The addition of important parameters, such as the width, the tortuosity, the branching of vessels, the presence of valves in lymphatics would have provided meaningful information which is missing to interpret the final data.

The last part is raising more interest as the mathematical model-imaging is used to investigate alterations of lymphatics in a model of lung fibrosis and interactions between tumor cells and blood vessels in a model of injected lung tumor cells. This later study is interesting and would deserve further trajectory analysis of tumor-vessel interactions, although the biological significance of the clustering remains limited.

The overall study appears to be not well connected with the reality and actual knowledge of vascular biology and neurovascular biology. The collaboration of vascular biologists and neurobiologists is apparently missing.

My recommendation is to submit this methodological study in a more specialized journal after shortening and improvement of the CUBIC imaging part, reinforcement of the set of data used for mathematical modeling of the vasculature, and collecting data demonstrating the biological significance and the advantage of this mathematical model-imaging for morphological analyses of vessels and their relationship with the microenvironment.

Reviewer #3:

Remarks to the Author:

A new analysis modality for vascular structures combining tissue clearing technology and

topological data analysis

Running title: Evaluation of vasculatures with CUBIC and TDA

Key words: 3D imaging, blood and lymphatic vasculatures, tissue-clearing technology, persistent homology

Key results: The paper "A new analysis modality for vascular structures combining tissue clearing technology and topological data analysis" presents a thorough study on vasculature networks (blood and lymphatic) measured by light sheet fluorescence technology on cleared whole organs from mouse models. Many 3D reconstructions are provided and demonstrate the high quality of this study. Interestingly the authors use a mathematical framework that was never used before for cleared organ 3D image analysis to extract geometric features and analyses the vasculature organization within whole organ.

Validity

In the current form the manuscript has some flaws that are described below. Once these points are addressed, the manuscript could be reconsidered for review.

Originality and significance

The method and results have a potential significance in the field of vascularization analysis and retains potential impact to pathological analysis of vascularization of intact cleared organs.

However, the originality of the data needs to be emphasized.

Data & methodology

Data and methodologies are not fully explained, and unfortunately lack important information.

Through the manuscript, there is a general lack of quantitative approach and most results do not support enough the author's conclusions. For instance, clear evidence that capillaries can be resolved and segmented are needed (e.g. the experimental resolution, which is completely different than the pixel resolution is never mentioned).

P. 10: please add a reference to the method section for the segmentation using iLastik. Please also state the number of classes that you used (only background and foreground or did you also add vessel boundaries?) and the classifier you used. What do you mean by "The structures of the lymphatic capillaries were extracted"? Are you referring to thresholding the probability map? If so, please indicate the probability threshold that you used or mention that you used a manual threshold by visually comparing in Imaris with the raw data and make it clear that the process is not automatic and might be biased. Also, it is of great importance to study the effect of this threshold. What is the outcome of changing the threshold by 1%, 5%, 10% on latter analyses with TDA?

P. 14 (and Fig. 7C): How are you able to train the machine learning framework to detect cancer cell? How do you decide which cell is cancerous when you label them? The accuracy of cancer cell detection is not mentioned, authors should at least provide the precision and recall on the training set to give the reader an idea of your method uncertainty.

Images are impressive and beautiful. Authors should highlight the spatial resolution and provide higher magnification images of the segmented features to evaluate the resolution limits.

Some figures are very difficult to interpret, and authors conclusions are not easily represented by the figures (e.g. the results indicated that both a-SMA+ mature blood vessels and VE-cad+ blood capillaries showed unique expansion patterns in the isocortex (Fig. 5E)).

Material and methods is missing information (e.g. age of animals, description of statistical analysis). Also, a table of clearing and labelling protocol, including concentration of antibodies and staining reagents and difference in incubation times for each organ can help the reproducibility of the experiments. Extraction of positive signal needs to be described in depth.

A benchmark between already existing vascular structure analysis is needed with the methods accuracy, uncertainty, and computing time. This would help the reader to better understand the novelty and advantage of using such analysis against others.

Authors should clearly state the limitation of the methodology. P. 18: you mention that your method needs to be improved. In what aspect should it be improved and what gain are you expecting when the improvements will be reached?

Appropriate use of statistics and treatment of uncertainties:

Number of repetitions of the experiment is missing in many figures. Description of statistical analysis is missing. In fig 5B error bar are missing.

Conclusions:

In the present form, conclusions are not enough supported by the data.

Suggested improvements:

A benchmark between already existing vascular structure analysis

Quantitative analysis needs to be added to reinforce the findings.

The manuscript is difficult to read, many expressions need to be rephrased (i.e. pp.9-10 "Present findings showed the utility of CUBIC 3D imaging for structural analysis of capillaries in vivo" . "The 3D images were so clear that we could capture the precise structure... ". Fig. 4 B-C: probability of what?

The manuscript needs to be re-written for a broad scientific public, the mathematical framework needs to be introduced and contextualized in the presented application. Authors need to clearly state the results and make a comparative analysis with other existing methodologies.

Figure legends need to be re-written, being quite repetitive and long and missing important information (i.e Fig 1B-G Are the images max intensity projection or a single plane? What is the Z-stack thickness, zoom factor; i.e Fig 3A, missing annotation on the part of the body described in the text); Fig. 5E: It is not clear to me how this plot is generated. What is multidimensional scaling (MDS)? Please explain this procedure or give a reference. Fig. 7A: please add the x (abscissa) unit for dist (is it in pixel?). In addition, figure legends include many details on the procedures that can be summarized in the methods part.

The paper would benefit from a figure where NPHH and TDA are applied on a probability map from the study (e.g. characterization of lymphatic vessels). The overview of the mathematical foundation is given in the methods section but without the context of this study it is not straightforward to understand by a broad audience. For example, it is difficult for me to understand how the plots from Fig. 5B were obtained using the proposed analysis, and what is actually pictured. In P. 15 you mention "To our best knowledge, this is the first study to conduct a structural analysis of mouse vasculatures using a combination of tissue-clearing technology and TDA" so a clear explanation of the application of TDA and PH seems mandatory. For example, you could take two vessel networks, compute the persistence diagram and apply the Wasserstein distance on both to illustrate what this framework captures. This would mean to add Fig. Supp. 8 in the main text with more details for the Wasserstein and MDS step. Please also state the requirements to perform such analysis (amount of RAM needed and time to compute).

Authors need to well reference the previous work, for instance CUBIC methodology has been already validated for vessel analysis (Nojima, et al Sci. Report 2017).

I advise the authors to refactor the paper in three parts:

1. Explaining TDA and NPHH analysis in the context of 3D vasculature imaging.
2. Results on their data and discussion about their findings.
3. Comparison with existing techniques with pros and cons.

Reviewer #4:

None

Responses to reviewers' concerns on Takahashi et al., "A new analysis modality for vascular structures combining tissue-clearing technology and topological data analysis."

We would like to thank all the reviewers for constructive comments to our current work. We have taken all these comments and suggestions into account in the revised version of our paper. We apologize that it took 11 months for us to revise the manuscript, because we have thoroughly checked our analytical methods, and extensively revised our manuscript.

Reviewer #1:

The manuscript describes a new approach for the analysis of vascular structures in mouse brains and other organs. For imaging established light sheet microscopy and clearing technologies are used with good results. The new twist in this study is the combination with machine learning techniques. Using this combination the authors could substantiate existing hypotheses and findings that were based on older techniques like standart histology. As this manuscript adds a new item to the toolbox of 3D imaging of cleared organs and animals it can be expected that future studies will build on this advance and from my point of view the study can be published

We are pleased that Reviewer#1 is overall positive and found our study interesting and novel as describing that "As this manuscript adds a new item to the toolbox of 3D imaging of cleared organs and animals it can be expected that future studies will build on this advance and from my point of view the study can be published".

Reviewer #2:

We appreciated the constructive comments of Reviewer #2 and addressed all the points.

The manuscript 'A new analysis modality for vascular structures combining tissue-clearing technology and topological data analysis' reports three-dimensional (3D) imaging studies of the blood and lymphatic vasculature in different organs, using the CUBIC tissue clearing method in combination with mathematical frameworks. The purpose of the study is 'to evaluate the usefulness of the tissue-clearing technology CUBIC for visualizing blood and lymphatic vessels in the adult mouse.

(1) The first part, corresponding to Figures 1-3, is dedicated to reproducing data already published by others in seminal papers on the CUBIC method, with the difference of using transgenic endothelial and lymphatic-specific reporter lines. The images shown are lacking definition and should be improved. The literature cited on the 3D-imaging of the blood and lymphatic vasculature is also incomplete. The conclusions drawn from these observations, such as 'these data support that CUBIC imaging is useful for visualization of blood vasculatures' or 'These data suggested organ- or area-specific structures of lymphatic vessels' may appear rather simplistic than providing new information on the organ-specific features of the vasculature.

We appreciate the reviewer's comments. We also recognized that several papers have already shown mouse blood/lymphatic vessels using CUBIC or other tissue-clearing methods (Miyawaki et al (2020), Di Giovanna et al (2017), Todorov et al (2020), Kirst et al (2020)). However, most of them mainly focused on mouse brain vasculatures and there was no report to show 3D images of other organs comprehensively, especially at whole mouse or organ level. As the reviewer pointed out, vascular structure was not studied in combination with CUBIC and VE-cad-tdTomato mice/Prox1-GFP mice. In addition, the simultaneous visualization of VE-cad/Prox1 or VE-cad/ α -SMA was not yet reported well. Therefore, we believe that our data is important to compare with the data of vasculatures stained with other markers and gives a new insight in this field.

Regarding VE-cad-tdTomato mice, it is known that VE-cad⁺ small blood capillaries of embryo or newborn baby can be monitored as 3D images without using tissue-clearing methods (Monvoisin et al (2006)). On the other hand, there are no comprehensive 3D data using adult mice, due to lack of imaging systems. Therefore, the importance of visualizing in adult mice using CUBIC is emphasized in our revised manuscript (p. 6).

In the case of Prox1-GFP mice, a beautiful work showed the brain lymphatic vessels using Prox1-GFP mice by vDISCO (Cai et al (2018)). Our data clearly supports their data. They focused especially on mouse adult brain, but other organs such as lung, kidney, heart, stomach were not yet analyzed well, suggesting the importance of Fig. 2.

We would like to emphasize that showing the 3D images of VE-cadherin-tdTomato and Prox1-GFP mice using CUBIC is necessary for future works. Following the reviewer's suggestions, we presented the former Figs. 1-3 in Figs. 1-2 and some figures were moved to Supplementary Figs. Instead of that, pixel classification, analysis with NHPP and PH was explained more precisely in Figs. 3-5. In addition, enlarged images were added in the revised main manuscript, and additional

references are listed (Liu et al (2019), Qi et al (2019), Messal et al (2021), Kirschnick et al (2021), Kostrikov et al (2021), Lagerweij (2017), Lugo et al (2017), Smith et al (2021)).

(2) The second part of the study aims at improving the quality of the vasculature analysis by using quantitative data extracted from the tiff images and processed by software and mathematical modeling. The procedure is applied to different organs, including the brain where it is used to cluster and distinguish arteries (SMA+) from other blood vessels (VE-cad+) based on parameters of intensity and geometric features. The procedure is only superficially explained, and the illustrating data (Figures 4C and 5C) are not convincing. It is hard to understand the benefit of this approach compared to other methods of quantifications, and the biological significance of the present clustering. The addition of important parameters, such as the width, the tortuosity, the branching of vessels, the presence of valves in lymphatics would have provided meaningful information which is missing to interpret the final data.

Thank you for the reviewer's comments. Following the reviewer's advice, we added more detailed explanations for each method and their importance in the biological context.

Using NHPP, not only the density of vasculatures, but also their distribution can be evaluated based on density data. In our original manuscript, we showed the distribution difference with only one value; however, we re-discussed and decided to show the data with separate parameters including strength (a) and directionalities (b_x , b_y , b_z). More specific procedures were explained with a new diagram in the revised manuscript (p. 11-12, see below, revised Fig. 4A). In addition, we also added 3 more independent samples (total $n = 4$) shown below (inserted in revised Supplementary Fig. 4). From the results, the density of α -SMA signal was higher especially in thalamus (TH), and the distribution of α -SMA signal was higher in isocortex (X-, Y-), pons (X-, Z-), midbrain (Y-) and medulla (X-) using this method (see below, revised Fig. 4B) (also see the responses to reviewer #3). These results were consistent with the previous data, showing that mature blood vessels exist on the surface of soft membrane under the arachnoid in the region from midbrain to brainstem (Xiong et al (2017), Dorr et al (2007)). The images of blood vessels were slightly faint, but analysis using NHPP clearly demonstrated the regional difference of cerebral blood vessels. Notably, analysis of blood vessels at the subarachnoid space may be clinically important because it is where bleeding is often observed.

On the other hand, utilization of “persistent homology (PH)” enables us to evaluate the vascular structure differences by making virtual circles and defining a new parameter “ r ”. Using this method, even if the volume of vasculatures, the number of branch points, and the length among branch points are the same but the structure is different, PH can detect its difference as explained in the revised manuscript (p. 12-13, see below, revised Fig. 5A), which is a strong point of this method. PH can make it possible for us to represent the structural differences identified by our eyes to the real numerical form. We believe utilization of this method in the biological area is very novel and meaningful.

We fully understand that the width and branching of vessels and the presence of valves can provide us meaningful information for vascular structures. However, these analyses with 3D images have just started to be reported and it is beyond our scope to explore these assessments in this current study. However, we totally agree that we should compare our new methods with the existing quantification methods and the comparison was added in Table in the revised manuscript (p. 19-20, Supplementary Table. 4) (also see the responses to reviewer #3 (8)).

Revised Fig. 4A

Overview of analysis procedure using CUBIC-Atlas and NHPP. First, signals of blood vessels were classified using ilastik software, then the brain anatomical information is added to them using ANTs and CUBIC-Atlas. To evaluate the intensities of blood vessels, non-homogeneous Poisson process (NHPP) was applied. In this method, strength (a) and directionalities (b_x , b_y , b_z) can be calculated using the pixel-based classified signals as the points. To compare the difference between α -SMA⁺ mature blood vessels and VE-cad⁺ blood capillaries, Fisher's exact test was used.

Revised Fig. 4B

Revised Supplementary Fig. 4A

Revised Supplementary Fig. 4B

Revised Fig. 5A

To extract the geometric features, persistent homology (PH), a main method of topological data analysis (TDA), is used. This method can evaluate structures by making virtual circles from the classified signals as a starting point. When the size of concentric circles was increased little by little, the larger circle will appear which connects them (birth time of circle). When the size of concentric circles become even larger, the appeared circles will disappear (dead time of circle). The radius of concentric circles at the birth time is defined as $r = r_b$, and the radius at the dead time is defined as $r = r_d$. These values (r_b and r_d) are plotted as persistent diagram. To compare two areas based on their features, the distance between two point clouds shown in PD was calculated by the Sliced Wasserstein kernel. Using the distance matrix obtained by calculating the distances between the point clouds in all pairs of areas by the Sliced Wasserstein Kernel as an input, the proximity between areas from a

geometric point of view is visualized by MDS.

(3) The last part is raising more interest as the mathematical model-imaging is used to investigate alterations of lymphatics in a model of lung fibrosis and interactions between tumor cells and blood vessels in a model of injected lung tumor cells. This later study is interesting and would deserve further trajectory analysis of tumor-vessel interactions, although the biological significance of the clustering remains limited.

Thank you for your comments on this data. We would like to present this finding in this paper because it shows the relationship between cancer and lymphatic vessels and it is attractive for readers of the paper.

(4) The overall study appears to be not well connected with the reality and actual knowledge of vascular biology and neurovascular biology. The collaboration of vascular biologists and neurobiologists is apparently missing.

My recommendation is to submit this methodological study in a more specialized journal after shortening and improvement of the CUBIC imaging part, reinforcement of the set of data used for mathematical modeling of the vasculature, and collecting data demonstrating the biological significance and the advantage of this mathematical model-imaging for morphological analyses of vessels and their relationship with the microenvironment.

We re-discussed these points with our collaborators, especially vascular biologists and neurobiologists, and added some descriptions about our results from the point of the biological significance (p. 11-14). Regarding the result of revised Fig. 4, α -SMA⁺ mature blood vessels surrounded midbrain to brainstem is well consistent with the previous observation (Xiong et al (2017), Dorr et al (2007)). In addition, PH detected well the representative structural features in ISO (penetrating branches from surface toward the center of the brain). To explain the result from NHPP and PH, we added some images (Fig. 4C and Fig. 5D-E). To fill the gap between mathematical models and the existing biology, we believe that this work provides us with a valuable insight in this field. Please also check the responses to Reviewer #3 ((6), (8)). We tried to explain the role of NHPP and PH in the biological context more clearly in the revised manuscript. We hope that the reviewer finds that this work is important.

Reviewer #3:

We appreciate the reviewer #3's valuable comments, which helped us to improve our manuscript. We are glad to hear that "Many 3D reconstructions are provided and demonstrate the high quality of this study. Interestingly the authors use a mathematical framework that was never used before for cleared organ 3D image analysis to extract geometric features and analyses the vasculature organization within whole organ." "The method and results have a potential significance in the field of vascularization analysis and retains potential impact to pathological analysis of vascularization of intact cleared organs." We addressed all concerns in full.

A new analysis modality for vascular structures combining tissue clearing technology and topological data analysis

Running title: Evaluation of vasculatures with CUBIC and TDA

Key words: 3D imaging, blood and lymphatic vasculatures, tissue-clearing technology, persistent homology

Key results: The paper "A new analysis modality for vascular structures combining tissue clearing technology and topological data analysis" presents a thorough study on vasculature networks (blood and lymphatic) measured by light sheet fluorescence technology on cleared whole organs from mouse models. Many 3D reconstructions are provided and demonstrate the high quality of this study. Interestingly the authors use a mathematical framework that was never used before for cleared organ 3D image analysis to extract geometric features and analyses the vasculature organization within whole organ.

Validity

In the current form the manuscript has some flaws that are described below. Once these points are addressed, the manuscript could be reconsidered for review.

Originality and significance

The method and results have a potential significance in the field of vascularization analysis and retains potential impact to pathological analysis of vascularization of intact cleared organs. However, the originality of the data needs to be emphasized.

Data & methodology

Data and methodologies are not fully explained, and unfortunately lack important information.

Through the manuscript, there is a general lack of quantitative approach and most results do not support enough the author's conclusions.

(1) For instance, clear evidence that capillaries can be resolved and segmented are needed (e.g. the experimental resolution, which is completely different than the pixel resolution is never mentioned).

Regarding this point, our CUBIC method can monitor the metastasis or vasculatures at the single-cell resolution with whole mouse/organ level. Our previous paper (Kubota, Takahashi et al (2017)) already showed this point with a nuclear staining. The same protocol including the light-sheet fluorescent microscopy was used for this work. We also addressed this point in the revised manuscript (p. 6).

(2) P. 10: please add a reference to the method section for the segmentation using iLastik. Please also state the number of classes that you used (only background and foreground or did you also add vessel boundaries?) and the classifier you used.

Thank you for the reviewer's comments. In our current study, two classes (target signal (vasculature or cancer cell) and background) were defined for the manual classification, and these were described in the revised manuscript (p. 26, Fig. 3A). Our recent paper (Kubota et al (2021), Commun Biol) clearly showed how to segmentate the imaging data using ilastik software, and demonstrated its validity. Following the reviewer's advice, we added another reference (Berg et al (2019)) and detailed classification procedures in Materials and Methods of our revised manuscript (p. 26).

(3) What do you mean by "The structures of the lymphatic capillaries were extracted"? Are you referring to thresholding the probability map? If so, please indicate the probability threshold that you used or mention that you used a manual threshold by visually comparing in Imaris with the raw data and make it clear that the process is not automatic and might be biased. Also, it is of great importance to study the effect of this threshold. What is the outcome of changing the threshold by 1%, 5%, 10% on latter analyses with TDA?

Thank you for the reviewer's suggestion, which indicated a very important point. In our current method, we used a manual threshold. The thresholds of classified signals were checked whether the threshold is correct or not by two different researchers. This description was added in the Materials and Methods part (p. 26). We think that this process could be automated in the future to reduce the bias caused by the manual threshold. We totally understood the reviewer's concern, which should be addressed in

the manuscript. Therefore, we analyzed one sample (Fig. 7 control#1, threshold_30%) as an example of TDA analysis of lung lymphatic vessels with different thresholds from 10-90% probability (see below) using persistent homology (PH). Interestingly, structural characteristics can be extracted with a 10-70% threshold very well. Focusing on the specific featured point (red arrow), that point was clear in a 10-70% threshold. The 90% threshold also can extract that point, but it is not clear compared to the ones with a 10-70% threshold, because of the low number of classified signals as the points for virtual circles. In addition, we compared these results with Fig. 6 and Fig. 7 (control only) (see below, Figures for reviewer 1). As a result, we can see the clear difference between the bleomycin group and the other groups at all thresholds 10-90%. Considering these results, the range of threshold is not narrow and the manual threshold can be used properly. Moreover, there is some visualization differences between samples, and manual threshold might be useful to deal with them. However, we cannot exclude the bias, so we decided to add other parameters such as the volume of classified signals in the revised manuscript (Fig. 6D and 7G). In addition, these Figures (see below) were inserted in Supplementary Figs. and this point was discussed in the revised manuscript as the limitation of this method (p. 13-14, 19-20).

Revised Supplementary Fig. 3

Revised Supplementary Fig. 3

Revised Supplementary Fig. 6

Figures for reviewer 1

(4) P. 14 (and Fig. 7C): How are you able to train the machine learning framework to detect cancer cell? How do you decide which cell is cancerous when you label them? The accuracy of cancer cell detection is not mentioned, authors should at least provide the precision and recall on the training set to give the reader an idea of your method uncertainty.

Our recent paper by Kubota et al (2021) already answered these questions. In that paper, we defined four segmentation parameters, i.e. cancer cells, autofluorescence, signal leakage along the Z-axis and background signal. As addressed above (2), we used two segmentation parameters, which are “cancer cells or vasculatures” and “other signals (= background)”. In addition, we compare the data with control samples, which are “no cancer injection for classification” side by side to decide which are target signals. In addition, from our previous experiments, we can speculate the rough number of pixels of cancer cells, and autofluorescence signals from bronchi can be excluded by manual segmentation.

(5) Images are impressive and beautiful. Authors should highlight the spatial resolution and provide higher magnification images of the segmented features to evaluate the resolution limits.

Thank you for the reviewer's suggestion. We added some higher magnification images of segmented features in the revised manuscript. These data were replaced with the old data in the original manuscript (Revised Fig. 3).

Revised Fig. 3B-D

(6) Some figures are very difficult to interpret, and authors conclusions are not easily represented by the figures (e.g. the results indicated that both α -SMA⁺ mature blood vessels and VE-cad⁺ blood capillaries showed unique expansion patterns in the isocortex (Fig. 5E)).

We appreciate and agree with the reviewer's comments. We added some representative magnified images of brain vessels, indicating the unique expansion patterns of α -SMA⁺ signals in the isocortex, midbrain, medulla and thalamus (see below, revised Fig. 4C and 5D-E), which were shown as a result of NHPP and PH. Probability cannot be visualized well with Imaris software, because of too many signals. Therefore, we would like to show the original data.

Revised Fig. 4C

Revised Fig. 5D and 5E

(7) Material and methods is missing information (e.g. age of animals, description of statistical analysis). Also, a table of clearing and labelling protocol, including concentration of antibodies and staining reagents and difference in incubation times for each organ can help the reproducibility of the experiments. Extraction of positive signal needs to be described in depth.

The ages of mice were written in each Figure legend and/or the Materials and Methods part. We added Supplementary Tables including labelling protocol, concentration of

antibodies, staining reagents and incubation time (see below, Supplementary Table 3 and 4). As addressed above, classification of target signals was addressed in the Materials and Methods part in the revised manuscript (p.26).

mouse body/organ	CUBIC-L	staining	decalcification	CUBIC-R
whole mouse	5-7 d	5-7 d	(-)	1-2 d
lung	1-3 d	3-4 d		
stomach				
intestine				
pancreas				
eye				
brain	2-4 d			
heart				
kidney				
spleen				
bone	1-3 d		1 week	
liver	5-7 d	5-7 d	(-)	

Supplementary Table 3. Time of CUBIC procedures

antibody	dilution
anti-GFP (conjugated with AlexaFluor 594 or 647)	1:100-1:200
anti-VEGFR3	1:100-1:200
anti- α -SMA-FITC	1:200-1:400

Supplementary Table 4. Antibody information

(8) A benchmark between already existing vascular structure analysis is needed with the methods accuracy, uncertainty, and computing time. This would help the reader to better understand the novelty and advantage of using such analysis against others.

Compared to the existing methodologies such as measuring the volume or the number of branches or width of vasculatures based on the 2D images, NHPP can provide us with an information that the density of each area and the directionality with 3D scale. PH makes us possible to evaluate the branching points and the length of each branch + distances between the branches with 3D images (see the responses to reviewer #2 (2)). Overall, our methods are more likely used at the whole organ level, which is important

for detecting the slight differences of organ conditions. We added the information including the comparison of the existing basic analyses in the revised manuscript (see below, Supplementary Table 1, p. 19-20).

Quantification methods	advantages	disadvantages
volume	Easy to measure	Lack of structure difference
branching point	Speculate developmental processes	(2D) Lack of spatial information (3D) Not established well
width	Speculate the directionality	
NHPP	Evaluate not only density but also directionalities (X-, Y-, Z-)	Quality of original images and classification process could partially affect results
TDA	Detect slight structural difference	

Supplementary Table 1. Comparison of NHPP and TH with existing methods

(9) Authors should clearly state the limitation of the methodology. P. 18: you mention that your method needs to be improved. In what aspect should it be improved and what gain are you expecting when the improvements will be reached?

Regarding limitations, the quality of original images could affect the results of NHPP and PH. Using a more superior light sheet fluorescence microscopy will improve this problem, and we are planning to do that in our future works. Another limitation is that the manual classification and deciding their threshold might have an influence on the result of NHPP or PH or further assessments. Ideally, these processes should be fully automated; however, we speculate that this process can be partially automated in the near future. In addition, parts of our methods like PH analysis needs hyper-spec computers, which also limit the application. These points were addressed in the Discussion part in the revised manuscript (p. 19-20).

(10) Appropriate use of statistics and treatment of uncertainties: Number of repetitions of the experiment is missing in many figures. Description of statistical analysis is missing. In fig 5B error bar are missing.

Regarding Fig. 5B, they are the result of Fisher's exact test. Therefore, it is not proper to add error bars. Instead of that, we added 3 more independent mouse samples in the revised manuscript Sup Fig. 4. These results suggested that the strength (a) and directionalities (b_x , b_y , b_z) of α -SMA⁺ blood vessels or VE-cad⁺ blood capillaries showed

the similar pattern throughout all 4 independent samples (Revised Supplementary Fig. 4, see the response to reviewer#2 (2)). Number of repetitions of each experiment was added in the Figure legends.

Conclusions:

In the present form, conclusions are not enough supported by the data.

Suggested improvements:

(11) A benchmark between already existing vascular structure analysis

Quantitative analysis needs to be added to reinforce the findings.

We added the total volume of classified signals as lymphatic vessels (revised Fig. 6D and Fig. 7D) and the discussion about the benchmarks of each existing method using Supplementary Table 1 (see above (8), p.19-20).

(12) The manuscript is difficult to read, many expression needs to be rephrased (i.e. pp.9-10 “Present findings showed the utility of CUBIC 3D imaging for structural analysis of capillaries in vivo” . “The 3D images were so clear that we could capture the precise structure... “. Fig. 4 B-C: probability of what?

Some phrases were modified to make the manuscript clearer by changing some words. “Extracted positive signals” may be misleading and should be avoided. Instead of that, “pixel classification” and “classified signals as lymphatic vessels” were used in the revised manuscript.

(13) The manuscript needs to be re-written for a broad scientific public, the mathematical framework needs to be introduced and contextualized in the presented application. Authors needs to clearly state the results and make a comparative analysis with other existing methodologies.

We already answered this question in (8).

(14) Figure legends need to be re-written, being quite repetitive and long and missing important information (i.e Fig 1B-G Are the images max intensity projection or a single plane? What is the Z-stack thickness, zoom factor; i.e Fig 3A, missing annotation on the part of the body described in the text);

Figure legends were re-written by adding some information such as Z-stack thickness and zoom factor. All data was captured with the merge of max intensity of left and right images. This description was added in the revised manuscript (p. 24).

(15) Fig. 5E: It is not clear to me how this plot is generated. What is multidimensional scaling (MDS)? Please explain this procedure or give a reference.

Thank you for pointing this out. Multidimensional Scaling (MDS) is a means to visualize the level of similarity of individual cases in a data set. In our analysis, MDS maps the information about the geometric distances between individuals using the sliced Wasserstein kernel, into an abstract Cartesian space. This description was added in the revised manuscript (p. 30-31) with the reference (Torgerson, W.S. (1952)).

(16) Fig. 7A: please add the x (abscissa) unit for dist (is it in pixel?). In addition, figure legends include many details on the procedures that can be summarized in the methods part.

Yes, we analyzed the distances between cancer metastasis and lymphatic vessels by pixel-based images. We added the details in the result part and Figure legends (p. 15, 43).

(17) The paper would benefit from a figure where NPHH and TDA are applied on a probability map from the study (e.g. characterization of lymphatic vessels). The overview of the mathematical foundation is given in the methods section but without the context of this study it is not straightforward to understand by a broad audience. For example, it is difficult for me to understand how the plots from Fig. 5B were obtained using the proposed analysis, and what is actually pictured.

We already answered this question in (6).

(18) In P. 15 you mention “To our best knowledge, this is the first study to conduct a structural analysis of mouse vasculatures using a combination of tissue-clearing technology and TDA” so a clear explanation of the application of TDA and PH seems mandatory. For example, you could take two vessel networks, compute the persistence diagram and apply the Wasserstein distance on both to illustrate what this framework

captures. This would mean to add Fig. Supp. 8 in the main text with more details for the Wasserstein and MDS step.

Thank you for the reviewer’s suggestion. We understand the reviewer’s point. To explain the application of TDA, an example should be demonstrated. However, we would say that showing the difference of two vascular network structures is not a good example, because we already know that two vascular structures are different, and the result of MDS would not give us beneficial information.

We totally agree that clearer explanation about TDA should be added in our manuscript. Instead of showing the example of TDA with two vascular vessels, we added the schematic diagram (Fig. 5A) and a more detailed description in our revised manuscript. In addition, the diagram of NHPP was also added in Fig. 4A (see the response to reviewer#2 (2)).

(19) Please also states the requirements to perform such analysis (amount of RAM needed and time to compute).

The requirements for this analysis including the amount of RAM and approximate time to compute were added as Supplementary Table 2.

Purpose	RAM	Time
ilastik	256 GB	Lung: 1-3 h/sample
Python	256 GB	Depending on the code e.g. file conversion: < 10-20 min/sample
Imaris software	For creating multi-color files or movies: 128 or 256 GB For viewing: 16 GB	Creating multi-color file: 10-20 min/sample
NHPP	128 GB	1 d/sample
Persistent homology	128 GB	10-20 min/sample

Supplementary Table 2. Information of computer specification

(20) Authors needs to well reference the previous work, for instance CUBIC methodology has been already validated for vessel analysis (Nojima, et al Sci. Report 2017).

We added some references including Nojima et al (2017), Liu et al (2019), Qi et al (2019), Messal et al (2021), Kirschnick et al (2021), Kostrikov et al (2021), Lagerweij (2017), Lugo et al (2017), Smith et al (2021) , which evaluate 3D vasculatures using tissue-clearing methods.

(21) I advise the authors to refactor the paper in three parts:

1. Explaining TDA and NPHH analysis in the context of 3D vasculature imaging.
2. Results on their data and discussion about their findings.
3. Comparison with existing techniques with pros and cons.

The meaning of PH and NHPP and their results were described especially in (6) and response to reviewer#2 (2), the comparison with existing methods was addressed in (8). In addition, all the points addressed above were appended in the revised manuscript.

Reviewers' Comments:

Reviewer #2:

Remarks to the Author:

The authors have answered my concerns with detail and improved the quality of the figures. In my opinion, the paper is now acceptable for publication.

Reviewer #3:

Remarks to the Author:

After carefully reading the author response and the new manuscript we think that it should not be accepted for publication in the current form. Although the substantial changes and the inclusion of new figures, statistics and references, the language is still unclear and the manuscript is hard to follow. Moreover, the authors do not provide quantitative comparisons with current methods. Below are a few critics that were not addressed by the authors and that could improve the manuscript:

- Authors only mention single-cell resolution when asked with the resolution. Please state the optical resolution.
- L540 "image analysis" should be "image analysis software". This paragraph is poorly written, but it is good that authors added a figure explaining the process (3A).
- Answer (4) is not satisfactory. It is not clear how the machine learning framework works, and the phrasing of authors answer is not helping.
- (5) authors only zoom with a factor 2, provide at least a factor 10 so readers can clearly see details.
- (6) provide 2D reconstruction video if the signals are not easily visualizable in 2D.
- (8) this is a qualitative comparison without any supporting material or reference.
- (9) The answer of authors suggests that as of now the presented method presents many drawbacks and features to be optimized. This again is not acceptable for publication. In summary the second resubmission has still major issues and the authors have missed some crucial points; therefore, we do not consider it eligible for publication.

Responses to reviewers' concerns on Takahashi et al., "A new analysis modality for vascular structures combining tissue-clearing technology and topological data analysis."

We would like to thank all the reviewers for constructive comments to our current work. We have taken all these comments and suggestions into account in the revised version of our paper.

Reviewer #2:

The authors have answered my concerns with detail and improved the quality of the figures. In my opinion, the paper is now acceptable for publication.

We are glad to hear that the reviewer #2 was positive and found our revised manuscript ready for publication.

Reviewer #3:

After carefully reading the author response and the new manuscript we think that it should not be accepted for publication in the current form. Although the substantial changes and the inclusion of new figures, statistics and references, the language is still unclear and the manuscript is hard to follow. Moreover, the authors do not provide quantitative comparisons with current methods.

Below are a few critics that were not addressed by the authors and that could improve the manuscript:

We appreciate the valuable comments of reviewer #3, which helped us to improve our manuscript. We have addressed all the concerns of reviewer #3.

- Authors only mention single-cell resolution when asked with the resolution. Please state the optical resolution.

The pixel resolution is partially described in the revised manuscript (p. 24-25). With our custom-made LSFM (Olympus), images were captured at $0.63 \times$ objective lens (NA = 0.15, working distance = 87 mm) with digital zoom from $0.8 \times$ to $6.3 \times$. This LSFM is equipped with sCMOS camera (NEO5.5, ANDOR) and their pixel number is 2560×2160 . Therefore, the pixel resolution is as follows. Images were captured with $z = 10 \mu\text{m}$ step and the pixel resolution(z) was $10 \mu\text{m}$. Some of the detailed descriptions were added in the 2nd revised manuscript.

zoom	pixel resolution	
	x	y
$\times 0.8$	12.9 μm	12.9 μm
$\times 1$	10.32 μm	10.32 μm
$\times 1.25$	8.25 μm	8.25 μm
$\times 1.6$	6.45 μm	6.45 μm
$\times 2.0$	5.16 μm	5.16 μm
$\times 2.5$	4.13 μm	4.13 μm
$\times 4.0$	2.58 μm	2.58 μm
$\times 6.3$	1.64 μm	1.64 μm

Regarding the lens (limit) resolution, we can calculate as below.

$$\text{Lens resolution} = 0.5 \times \text{observation wavelength } (\mu\text{m})/\text{NA}$$

$$\text{Observation wavelength} = 0.550, \text{ NA} = 0.15$$

$$\text{Lens resolution} = 0.5 \times 0.550/0.15 = 1.83 \mu\text{m}$$

- L540 “image analysis” should be “image analysis software”. This paragraph is poorly written, but it is good that authors added a figure explaining the process (3A).
- Answer (4) is not satisfactory. It is not clear how the machine learning framework works, and the phrasing of authors answer is not helping.

We have rewritten the paragraph (p. 26-28) and included more detailed explanation of how the machine learning framework works using ilastik software. As shown in the revised Fig. 3a, ilastik software learns from manual labeling by the users (Mano et al (2021), Kubota et al (2021)). The original ilastik paper (Sommer et al (2011)) showed that a random forest classifier was used in the learning step, in which each pixel’s neighborhood was characterized by a set of generic (nonlinear) features. In the training phase, users can fine-tune by interactively providing new labels. Therefore, we repeated the training until the prediction map looked correct. We also annotated several samples with labeling to increase the robustness.

- (5) authors only zoom with a factor 2, provide at least a factor 10 so readers can clearly see details.

Following the reviewer’s advice, we have added more magnified images (Revised Figures 3b and 3d).

Revised Fig. 3b

Revised Fig. 3d

•(6) provide 2D reconstruction video if the signals are not easily visualizable in 2D. Thank you for the suggestion. We have added a 2D reconstruction video of brain, which allows us to get the difference between the α -SMA⁺ signals and VE-cad⁺ signals (Revised Supplementary Movie 3).

• (8) this is a qualitative comparison without any supporting material or reference. Following the reviewer's suggestion, we decided to quantify other existing parameters such as branching points. Using "TubeMap" (Krist et al, (2020)), the numbers of branching/end points (=vertex), the lengths between them (=edge length), and their radii (=vertex radius and edge radius) can be calculated in 3D images as shown below (Revised Supplementary Fig. 6a). The result of brain vessels showed that the vertex and edge numbers of VE-cad⁺ blood vessels were higher than those of α -SMA⁺ blood vessels (Revised Supplementary Fig. 6b). The vertex and edge numbers in the isocortex were prominently higher compared to the other brain areas (Supplementary Fig. 6b). Regarding the edge radii, those of α -SMA⁺ blood vessels in the isocortex area were larger than those of VE-cad⁺ blood vessels. These results supported our TDA and NHPP data and showed that the structure of blood vessels in the isocortex area is unique compared to the other brain regions.

Revised Supplementary Fig. 6a

Revised Supplementary Fig. 6b

Although the TubeMap was developed for brain vasculature analysis, we also applied this to our data of lung lymphatic vessel (Fig. 6 and Fig.7). The numbers of vertex in bleomycin-treated lungs were smaller than those in control or saline-treated lungs (Revised Fig. 6c). In addition, the edge lengths and the edge radii were also smaller in bleomycin-treated lungs (Revised Fig. 6c). These data were consistent with our TDA analysis data. Regarding B16F10 lung metastasis model, the edge lengths and radii of lymphatic vessels on Day 4 were smaller than those in control (Revised Fig. 7f). These data also supported our TDA data. The edge lengths on Day 10 were also smaller, from which we can speculate that the B16F10 colonies existed between the lymphatic vessels and disturbed their signals.

In this way, use of other existing parameters allowed us to speculate the vascular structure differences and the results were consistent with our TDA/NHPP analysis data. However, we still believe that TDA and NHPP are useful in obtaining the structural difference comprehensively including these existing parameters.

Revised Fig. 6c

Revised Fig. 7f

•(9) The answer of authors suggests that as of now the presented method presents many drawbacks and features to be optimized. This again is not acceptable for publication. In summary the second resubmission has still major issues and the authors have missed some crucial points; therefore, we do not consider it eligible for publication.

Compared to other existing quantification methods, our new methods allow us to speculate differences in overall structure from the "shape" of the data, while not depending on the choice of metrics, such as length, branching points, and width, and providing stability against noise (Revised Supplementary Table 1, below). Therefore, we can get more comprehensive understanding of differences in vascular structure. We are aware that our methods have some limitations; however, these are also problems with other existing quantification methods. We believe that these will be overcome in the future, and we still believe that TDA and NHPP are useful for 3D vasculature structure analysis.

Revised Supplementary Table 1

Methods	Structural information obtained	Missing structural information	Other limitations	
TubeMap	Branching/end points, length and radius of branches	Spatial information		Quality of original images and classification process may partially affect results
NHPP	Signal densities and their directionalities (X-, Y-, Z-)	Structural parameters such as branching points and radius of branches	High computational cost	
TDA	Overall structure from the "shape" of the data, while not depending on the choice of metrics and providing stability against noise			

Reviewers' Comments:

Reviewer #3:

Remarks to the Author:

The authors improved the quality of the manuscript by benchmarking the tool with other current methods and providing enough details for the reproducibility of the experiments. We consider the manuscript acceptable for publication.

Responses to reviewers' concerns on Takahashi et al., "A new analysis modality for vascular structures combining tissue- clearing technology and topological data analysis."

Reviewer #3:

The authors improved the quality of the manuscript by benchmarking the tool with other current methods and providing enough details for the reproducibility of the experiments. We consider the manuscript acceptable for publication.

We are really glad that the reviewer #3 is positive and thinks our manuscript ready for publication. We appreciate many valuable comments from the reviewer #3.

We have thoroughly checked the figures again, and found out that Supplementary Figure 4 showed "b_x" values instead of "a" values. Therefore, we revised this Figure with a new graph. In addition, we revised Supplementary Figure 2e with other images so that readers can easily understand our findings. These changes did not affect our conclusions.